# Distributional Monte-Carlo Planning with Thompson Sampling in Stochastic Environments

## Abstract

We focus on a class of reinforcement learning algorithms, Monte-Carlo Tree Search (MCTS), in stochastic settings. While recent advancements combining MCTS with deep learning have excelled in deterministic environments, they face challenges in highly stochastic settings, leading to suboptimal action choices and decreased performance. Distributional Reinforcement Learning (RL) addresses these challenges by extending the traditional Bellman equation to consider value distributions instead of a single mean value, showing promising results in Deep Q Learning. In this paper, we bring the concept of Distributional RL to MCTS, focusing on modeling value functions as categorical and particle distributions. Consequently, we propose two novel algorithms: Categorical Thompson Sampling for MCTS (CATS), which uses categorical distributions for Q values, and Particle Thompson Sampling for MCTS (PATS), which models Q values with particle-based distributions. Both algorithms employ Thompson Sampling to handle action selection randomness. Our contributions are threefold: We introduce a distributional framework for Monte-Carlo Planning to model uncertainty in return estimation. We prove the effectiveness of our algorithms by achieving a non-asymptotic problem-dependent upper bound on simple regret of order $O(n^{-1})$, where $n$ is the number of trajectories. We provide empirical evidence demonstrating the efficacy of our approach compared to baselines in both stochastic and deterministic environments.

## 1 Introduction

*Online planning* in Markov decision processes (MDPs) involves making real-time decisions based on the current state of the environment. It requires balancing exploration and exploitation while handling uncertainty and partial observability. Monte Carlo Tree Search (MCTS) is a highly effective online planning method for tackling complex MDPs. MCTS has shown impressive performance in various tasks, including traditional board games like Chess and Go, video games, and real-world challenges. Notable successes include advancements in Chess (35) and Go (34; 36; 30), video game strategy (28), robot assembly (16), robot path planning (15; 13), and autonomous driving (24).

Despite these achievements, current MCTS methods are primarily effective in deterministic environments, often overlooking the significant impact of randomness in real-world scenarios. In highly stochastic and partially observable environments, conventional MCTS approaches face substantial challenges due to widespread randomness and limited observability. This leads to compromised value estimates, suboptimal decisions, and diminished overall performance. Therefore, there is a clear need for improved methods capable of navigating the complexities of randomness and partial observability in value estimation.

We now review related works to understand the advancements and limitations in these areas.

**Related work** In MCTS, value estimation methods and action selection rules are critical factors for algorithm performance. Traditional value estimation methods, such as using empirical average mean

for value backup as in the Upper Confidence bounds applied to Trees method (UCT) (21), suffer from underestimation of optimal values while maximum backup suffers from overestimation of optimal values (9). The power mean estimator (12) offers a balanced solution by computing a mean between the average and maximum values. In our approach, we also use power mean for value operator as each V node stores the power mean of empirical means of succeeding Q-value nodes, eliminating the need for V to be modeled as a distribution.

For action selection in MCTS, strategies from Multi-Armed Bandits (MAB) are commonly employed. For instance, UCT extends the UCB1 strategy from bandits to the tree by computing confidence intervals at each step. However, original UCT's performance is hindered by the incorrect choice of logarithmic bonus constant (32). Shah et al. (32) propose an adapted version of UCT incorporating a polynomial bonus term instead of the "logarithmic" bonus term in UCT and show the non-asymtotic convergence of rate $O(n^{-1/2})$, with $n$ is the number of rollout trajectories. On the other hand, our method improves over this rate with theoretical guarantee of $O(n^{-1})$. Although Thompson sampling has been less explored in MCTS, some approaches like those by Bai et al. (1) and Bai et al. (2) incorporate it for exploration. However, these methods lack convergence rate analysis. Furthermore, in the article Bai et al. (1), authors model value functions as a mixture of Normal distributions, which may lack the generality of complex real-world scenarios. Our approach adopts Thompson sampling for action selection but introduces a novelty by modeling the uncertainty of action value estimates over the tree as arbitrary categorical and particle-based distributions. This modification enhances our ability to handle more generality in highly stochastic environments effectively.

*Entropy regularization* techniques in RL modify value and action selection functions to balance exploration and exploitation, leading to improved value estimation (25; 17; 31; 18). Several works have applied these techniques in MCTS. Maximum Entropy Tree Search (MENTS) (40) emphasizes exploration by integrating MCTS with maximum entropy policy optimization. MENTS aims to maximize cumulative rewards and policy entropy concurrently, regulated by a temperature parameter. Dam et al. (14) extend MENTS by incorporating Relative and Tsallis entropy, leading to the RENTS and TENTS algorithms. However, the effectiveness of MENTS/RENTS/TENTS hinges on the temperature parameter, which may impede convergence. Furthermore, the value estimation converges exponentially to the regularized value not the optimal one. In contrast, Painter et al. (27) utilize a similar action selection approach but employ a maximum backup operator for value estimation. Although their method exhibits exponential decay of simple regret, it heavily relies on the sensitivity of the temperature parameter for Boltzmann Exploration, limiting its practicality.

*Distributional Reinforcement Learning* (RL) (6; 11; 22) addresses the randomness of the value estimation by introducing a distributional perspective to the traditional Bellman equation. This approach views the value function as a distribution rather than a single mean, providing a comprehensive understanding of uncertainties in rewards and the stochasticity from environments. Through discretization (26), parameterization (6), and quantization (10), it allows for efficient and effective approximation of value distributions, leading to improved performance in various RL tasks. However, these results are only for *learning* not for *planning*.

**Outline and contribution** In this work, we integrate the distributional approach from reinforcement learning (RL) into the *planning* framework to tackle the challenges of planning in stochastic environments. We focus on modeling value functions as categorical and particle distributions. Consequently, we propose two novel algorithms: Categorical Thompson Sampling for MCTS (CATS) and Particle Thompson Sampling for MCTS (PATS). CATS represents each Q value function as a categorical distribution and uses Thompson Sampling for action selection to manage uncertainty. PATS models each Q value function with a particle-based distribution, using a nuanced Thompson Sampling approach to handle action selection randomness.

Our contributions are threefold:

(i)    In section 3, we introduce a distributional framework for *planning* to model uncertainty in return estimation, enhancing the robustness of value estimation in stochastic environments.
(ii)   In section 4 Theorem 5 and Theorem 6, we prove the effectiveness of our algorithms by achieving a non-asymptotic problem-dependent upper bound on simple regret of $O(n^{-1})$, which significantly improves upon the current state-of-the-art theoretical analysis of regret, previously established at $O(n^{-1/2})$ by Shah et al. (33).

92     (iii)   In section 5, we provide comprehensive empirical evidence demonstrating the efficacy of
93             our approach compared to baselines, showcasing competitive performance in stochastic
94             settings and the Atari benchmark.

95   In the next section, we describe the problem setting addressed in this paper.

## 96   2   Setting

97   In our study, We address the dynamics of an agent navigating an infinite-horizon discounted Markov
98   decision process (MDP), defined formally as $\mathcal{M} = \langle \mathcal{S}, \mathcal{A}, \mathcal{R}, \mathcal{P}, \gamma \rangle$. Here, $\mathcal{S}$ represents the state
99   space, $\mathcal{A}$ denotes the set of actions, and $\mathcal{R}$ quantifies the Reward function of the MDP ($\mathcal{R} : \mathcal{S} \times$
100  $\mathcal{A} \times \mathcal{S} \to \mathbb{R}$). Transition dynamics are governed by $\mathcal{P}(\mathcal{S} \times \mathcal{A} \to \mathcal{S})$, with $\gamma \in (0, 1]$ as the discount
101  factor. The agent interacts with the environment via a policy $\pi \in \Pi : \mathcal{S} \to \mathcal{A}$, guiding action
102  selection based on observed states. This yields an action-value function $Q^\pi$, indicating the expected
103  cumulative discounted reward from a state-action pair under $\pi$. The agent seeks the optimal policy
104  maximizing the action-value function, adhering to the Bellman equation (7), given by $Q(s, a) \triangleq$
105  $\int_{\mathcal{S}} \mathcal{P}(s'|s, a)[\mathcal{R}(s, a, s') + \gamma \max_{a'} Q(s', a')]ds$ for all states $s$ and actions $a$. Upon acquiring the
106  optimal action-value function, we derive the optimal value function $V(s) \triangleq \max_{a \in \mathcal{A}} Q(s, a)$ for all
107  states $s$ in $\mathcal{S}$.

108  **Monte-Carlo tree search** (MCTS) (20; 8) is a planning approach for complex Markov decision
109  processes (MDPs). It employs an iterative approach:
110  _Selection_: It begins by selecting an action using a specified strategy, followed by executing this action
111  through Monte Carlo simulation.
112  _Expansion_: Subsequently, it assesses the resulting state, either by recursively evaluating if it already
113  exists in the search tree or by inserting it into the tree.
114  _Simulation_: Or employing a rollout policy via simulations. This iterative process continues until
115  certain termination criteria are met, allowing traversal through the search tree.
116  _Backpropagation_: Finally, the outcomes of the simulations are propagated backward through the
117  chosen nodes to update their statistical metrics.

118  **Simple Regret** An MCTS algorithm dynamically gathers trajectories within an MDP starting from
119  an initial state $s_0$. After processing $t$ trajectories, it provides two outputs:
120   •   $\widehat{a}_t$, a guess for the best action to take at state $s_0$
121   •   $\widehat{V}_t(s_0)$ an estimator of the optimal value in $s_0$,
122  where $s_0$ is the state at the root node. The algorithm's performance can be assessed by its convergence
123  rate $r(t)$ of the simple regret, formulated as:

$$\mathbb{E}\left[R(s_0, t)\right] = \mathbb{E}\left[V^\star(s_0) - \widehat{V}_t(s_0)\right] \le r(t),$$

124  Here, $R(s_0, t) = V^\star(s_0) - \widehat{V}_t(s_0)$ is the simple regret of the algorithm at the root node with $V^\star(s_0)$
125  representing the optimal value at state $s_0$.

126  In this article, we analyze an MCTS algorithm employing a maximal planning horizon $H$ and
127  a playout policy $\pi_0$ with value $V_0$. We define $\widetilde{V}(s_H) = V_0(s_H)$ recursively as follows: for all
128  $h \le H - 1$,

$$\widetilde{Q}(s_h, a) = r(s_h, a) + \gamma \sum_{s_{h+1} \in \mathcal{A}_{s_h}} \mathbb{P}(s_{h+1}|s_h, a)\widetilde{V}(s_{h+1}), \widetilde{V}(s_h) = \max_a \widetilde{Q}(s_h, a), \qquad (1)$$

129  where $r(s_h, a)$ defined formally as the mean intermediate reward at state $s_h$ after taking action $a$.
130  The primary objective of an MCTS algorithm is to estimate a tied rate $r(t)$ by constructing estimates
131  of $\widetilde{Q}(s_h, a)$ and $\widetilde{V}(s_h)$ to ultimately estimate $\widetilde{Q}(s_0, a)$ and consequently $Q^\star(s_0, a)$. In practical im-
132  plementations of the MCTS algorithm, the maximal depth $H$ can sometimes be set to $+\infty$. However,
133  for theoretical analysis, the maximal depth $H$ is crucial as we will analyze the algorithm that always
134  collects trajectories of length H.

135  **Distributional Reinforcement Learning** The mathematical framework used in reinforcement learn-
136  ing is based on the Bellman equation (37), which aims to find an agent to maximize the expected
137  utility Q value. However, the single expected value function cannot encapsulate the stochasticity in
138  the reward function and the dynamic of the environments. Recently, in the article (5), authors shed
139  light on the distributional perspective of the Bellman equation by modeling each Q value function as
140  a distribution instead of a single expected value. The main objective is to study the random return $\mathcal{Q}$
141  at the state $s$, action $a$, and is defined recursively as

$$\mathcal{Q}(s, a) \stackrel{D}{=} \mathcal{X}(s, a) + \gamma \mathcal{Q}(s', a'), \mathcal{V}(s') \stackrel{D}{=} \mathbb{E}_\pi \mathcal{Q}(s', \pi(\cdot|s')), \qquad (2)$$

where $\mathcal{X}(s, a)$ is the reward distribution at the state $s$, action $a$, $\mathcal{Q}(s, a)$ is the Q value distribution at state $s$, action $a$, and $\mathcal{Q}(s^{'}, a^{'})$ is the Q value distribution at state $s^{'}$, action $a^{'}$. $s^{'}$ distributed according to $\mathbb{P}(\cdot|s, a)$, $a^{'}$ distributed according to a policy $\pi(\cdot|s^{'})$. $A \overset{D}{=} B$ denotes that two random variables $A$ and $B$ have equal probability laws.

This distributional approach offers a deeper understanding of uncertainty and variability, especially in complex, stochastic systems where traditional expected value representations may fail to capture the true dynamics of the problem. which has been successfully used in Deep Q Learning (5).

**Categorical Value Distribution** Based on the distributional Bellman equation, In the article (5), authors approximate the Q value distribution $\mathcal{Q}(s, a)$ as a discrete categorical distribution parametrized by $N \in \mathbb{N}$, which denotes the number of atoms (N+1) at fixed-sized locations. This method effectively divides the Q value function into a set of equally spaced atoms $z_i(s, a) = Q_{min} + i \triangle z : 0 \le i \le N$, where $Q_{min}$ and $Q_{max}$ are respectively the minimum and maximum values at state $s$, action $a$. The size of each atom is set as $\triangle z := \frac{Q_{max} - Q_{min}}{N}$.

This discrete distribution approach is highly expressive and computationally efficient, making it ideal for practical applications. For instance, in the article (5), authors successfully used this representation in Deep Q Learning (C51), showing promising results in several Atari games. In the next section, we demonstrate how to apply this idea to MCTS.

# 3 Distributional Thompson Sampling in Tree Search

In this section, we introduce two novel distributional approaches for MCTS based on Thompson sampling. The first method represents each Q-value node as a categorical distribution, while the second uses particle-based distributions for greater flexibility. Both methods integrate Thompson sampling for improved exploration and performance.

## 3.1 Distributional Monte-Carlo Tree Search

We leverage the success of distributional reinforcement learning (4; 3; 6) and apply this concept to MCTS. In MCTS, there are two types of nodes: V-nodes and Q-value nodes. Instead of treating each V value and Q value as a single expected value, we model these functions as distributions.

Based on equation (2), we can derive

$$\mathcal{Q}(s, a) \overset{D}{=} \mathcal{X}(s, a) + \gamma \mathcal{V}(s^{'}), \mathcal{V}(s^{'}) \overset{D}{=} \sum_{a^{'} \sim \bar{\pi}(.|s^{'})} \mathcal{Q}(s^{'}, a^{'}), \tag{3}$$

with $s^{'} \sim \mathbb{P}(\cdot|s, a)$, where $\bar{\pi}(.|s^{'})$ is formally defined as the tree policy at state $s'$. We can model any Q distribution with equal law distributed as the sum of the distributions of the next reward and the Q distributions of the next states actions. We further model each V distribution, having equal probability law to the expectation of the chosen policy of the next Q-value distributions (3).

Our method follows the same four basic steps of MCTS but is different in Value Backup and Action selection steps. We introduce two distinct methodologies: categorical-based and particle-based. In the categorical based approach, we parameterize each V value and Q value function in the tree as a categorical distribution. In contrast, in the particle-based approach, we model each value distribution as a set of sampling particles, representing the values observed during the tree planning. We provide a detailed explanation for the value backup and action selection of each method in the next section.

## 3.2 Value Backup

In this work, we employ two approaches to represent the Q value distribution.

**Categorical distribution**: we represent each node in the tree as a categorical distribution. In each Q-value node, we: (1) store the empirical mean value of that Q-value node (same as in UCT), and (2) maintain a categorical distribution of the Q value function. To define a categorical distribution Q function, we require three essential pieces of information:

- The number of atoms $(N + 1)$: We choose a consistent number of atoms $(N + 1)$ that remains the same for all Q distributions along the tree.
- Minimum and maximum values (*min* and *max*): Each node in the tree may have different ranges for its minimum $(Q_{min})$[1] and maximum $(Q_{max})$ values, depending on its state/action in the environment. When a new Q-value node is added to the tree, we initially set $Q_{min}$ to 0 (assuming we have scaled the reward range to [0, R]) and initialize $Q_{max}$ to a small

---

[1]Since reward is scaled in $[0, R]$, $Q_{min}$ is not updated in our setup.

| **Algorithm 1** CATS | **Algorithm 2** PATS |
|---|---|

**SelectAction** $(s_h)$ *(Sec 3.2)*
  **for** $a \in [A]$ **do**
    $L(s_h, a) \sim \text{Dir}(\alpha^0(s_h, a), \ldots, \alpha^N(s_h, a))$
    $\overline{\phi}(s_h, a) = [z_0(s_h, a), \ldots, z_N(s_h, a)]^\top L(s_h, a)$
  $a = \arg\max_a \{\overline{\phi}(s_h, a)\}$
  **return** $a$

**SimulateV** $(s_h, t)$ *(Sec 3.2)*
  $a = \text{SelectAction}(s_h)$
  $\text{SimulateQ}(s_h, a, t)$
  $T_{s_h}(t) = T_{s_h}(t) + 1$
  $\widehat{Q}(s_h, a) = \sum_i z_i(s_h, a) p_i(s_h, a)$
  $\widehat{V}(s_h) = \left( \sum_a \frac{T_{s_h,a}(t)}{T_{s_h}(t)} \widehat{Q}^p(s_h, a) \right)^{\frac{1}{p}}$

**SimulateQ** $(s_h, a, t)$ *(Sec 3.2)*
  $s_{h+1} \sim \mathbb{P}(\cdot|s_h, a), r_t(s_h, a) \sim \mathcal{R}(s_h, a, s_{h+1})$
  **if** *Node* $s_{h+1}$ *not expanded* **then**
    $\text{Rollout}(s_{h+1})$
  **else**
    $\text{SimulateV}(s_{h+1}, t)$
  $T_{s_h,a}(t) = T_{s_h,a}(t) + 1$
  $\overline{Q}_t(s_h, a) = r_t(s_h, a) + \gamma \widehat{V}(s_{h+1})$
  **if** $\overline{Q}_t(s_h, a) \notin [Q_{\min}(s_h, a), Q_{\max}(s_h, a)]$ **then**
    $Q_{\max}(s_h, a) = \max\{\overline{Q}_t(s_h, a), Q_{\max}(s_h, a)\}$
    $Q_{\min}(s_h, a) = \min\{\overline{Q}_t(s_h, a), Q_{\min}(s_h, a)\}$
    $\triangle z = \frac{Q_{max} - Q_{min}}{N}$
    $z_i(s_h, a) = Q_{min} + i \triangle z : 0 \leq i \leq N$
  $\text{Update } p(s_h, a) = [p_0(s_h, a), \ldots, p_N(s_h, a)]$

**SelectAction** $(s_h)$ *(Sec 3.2)*
  **for** $a \in [A]$ **do**
    $L(s_h, a) \sim \text{Dir}(\alpha(s_h, a))$
    $\overline{\phi}(s_h, a) = \mathcal{S}(s_h, a)^\top L(s_h, a)$
  $a = \arg\max_a \{\overline{\phi}(s_h, a)\}$
  **return** $a$

**SimulateV** $(s_h, t)$ *(Sec 3.2)*
  $a = \text{SelectAction}(s_h)$
  $\text{SimulateQ}(s_h, a, t)$
  $T_s(t) = T_s(t) + 1$
  $\widehat{Q}(s_h, a) = \sum \alpha_t(s_h, a) \overline{Q}_t(s_h, a)$
  $\widehat{V}(s_h) = \left( \sum_a \frac{T_{s_h,a}(t)}{T_{s_h}(t)} \widehat{Q}^p(s, a) \right)^{\frac{1}{p}}$

**SimulateQ** $(s_h, a, t)$ *(Sec 3.2)*
  $s_{h+1} \sim \mathbb{P}(\cdot|s_h, a), r_t(s_h, a) \sim \mathcal{R}(s_h, a, s_{h+1})$
  **if** *Node* $s_{h+1}$ *not expanded* **then**
    $\text{Rollout}(s_{h+1})$
  **else**
    $\text{SimulateV}(s_{h+1}, t)$
  $T_{s_h,a}(t) = T_{s_h,a}(t) + 1$
  $\overline{Q}_t(s_h, a) = r_t(s_h, a) + \gamma \widehat{V}(s_{h+1})$
  **if** $\overline{Q}_t(s_h, a) \in \{\mathcal{S}(s_h, a)\}$ **then**
    $\alpha_t(s_h, a) \mathrel{+}= 1$ // $\alpha_t(s_h, a)$ : weight of $\overline{Q}_t(s_h, a)$
  **else**
    $\mathcal{S}(s_h, a) := (\mathcal{S}(s_h, a), \overline{Q}_t(s_h, a))$
    $\alpha(s_h, a) := (\alpha(s_h, a), 1)$

Figure 1: Comparing CATS (left) and PATS (right) The main distinction is in the Q value function backup(**SimulateQ**) and action selection function (**SelectAction**); the two methods are identical in other procedures. In CATS, we init $(\alpha^0(s, a), \ldots, \alpha^N(s, a)) = (1, \ldots, 1)$ and in PATS, $\mathcal{S}(s, a) = (1), \alpha(s, a) = (\varnothing)$ for each $s, a$.

number, e.g., $Q_{max} = 0.001$. Since the min and max values are unknown, we start with a small range, that will get updated accordingly to the scale of the observed values.

- Probabilistic parameterization: The probability of each atom $(p_i(s, a))$ is determined based on the visitation count ratio. In detail, each atom stores statistical information about the visitation count, and the probability of that atom will be calculated as the visitation count divide with the total visitation count of that Q-value node. When we backpropagate the $r_t(s, a) + \gamma \widehat{V}_t(s')$ value to a specific node, we identify the atom whose value range includes the $r_t(s, a) + \gamma \widehat{V}_t(s')$ value. At this point, we increase its visitation count.

Additionally, as we backpropagate Monte-Carlo Q values over time, we empirically adjust the $Q_{min}$ and $Q_{max}$ values to account for the dynamic range of Q values observed in the tree. This dynamic scaling ensures that the atom locations are effectively rescaled to adapt to the changing conditions. This representation method allows us to encapsulate the knowledge gained through exploration in the form of categorical distributions, which helps in making informed decisions during the tree search.

**Paricle based distribution**: We represent each Q value distribution as a collection of sampling particles, which encapsulate the observed values during tree planning. Initially, we maintain an empty set of particles for the Q value distribution, denoted as $\mathcal{S}(s, a)$. At time step $t$, upon receiving an intermediate reward $\overline{Q}_t(s, a) = r_t(s, a) + \gamma \widehat{V}_t(s')$, with $s' \sim \mathbb{P}(\cdot|s, a)$, we add $\overline{Q}_t(s, a)$ to the set $\mathcal{S}(s, a)$ if the particle does not already exist within it. If the particle $\overline{Q}_t(s, a)$ already exists in $\mathcal{S}(s, a)$, we increase the visitation count ratio associated with that particle.

**Value function:** The Q-value node is crucial in the tree because its representation influences action selection, as detailed in the next section. We now discuss modeling each V-value node. The V-value distribution is based on the expected outcomes of the chosen policy and the subsequent Q-distributions. Thus, the mean of the V-function corresponds to the tree policy's expectation of the means of all

succeeding Q-value nodes. The common approach is to use empirical average mean for the value backup, as in UCT (21). However, this approach underestimates the optimal value, while using the maximum value overestimates it (9). The power mean estimator (12) provides a balanced solution, falling between the average and maximum values. In our methods, each V node stores the power mean of the empirical means of all succeeding Q-value nodes, eliminating the need to model V as a distribution.

$$\widehat{V}(s) = \left( \sum_a \frac{T_{s,a}(n)}{T_s(n)} \widehat{Q}^p(s,a) \right)^{\frac{1}{p}}, p \geq 1,$$

where $T_s(n), T_{s,a}(n)$ are the number of visitations at $s$ and $s, a$ at timestep $n$ respectively. Next, we show how to select actions in the tree based on the categorical distribution of Q-value nodes.

## 3.3 Action Selection

Thompson sampling has shown promising results in real bandit scenarios due to the randomness of action selection. Taking advantage of the established categorical based distribution and particle based distribution, we use the Thompson sampling method for action selection. We maintain a Dirichlet distribution of parameter of the Q value distribution. We denote the Dirichlet distribution of parameters $(\alpha^0, \alpha^1, \ldots, \alpha^N)$ by $\text{Dir}(\alpha^0, \alpha^1, \ldots, \alpha^N)$, whose density function is given by $\frac{\Gamma(\sum_{i=0}^N \alpha^i)}{\Pi_{i=0}^N \Gamma(\alpha^i)} \Pi_{i=0}^N x_i^{\alpha^i - 1}$ for $(x_0, \ldots, x_N) \in [0,1]^{N+1}$ such that $\sum_{i=0}^N x_i = 1$.

**Categorical distribution**: The probability mass function of the discrete categorical distribution at each Q-value node at state $s$, action $a$: $p(s,a) = [p_0(s,a), p_1(s,a), \ldots, p_N(s,a)]$, where $p_i(s,a)$ represents the probability of selecting the $i$-th atom $z_i(s,a)$, $N+1$ is the number of atoms. We maintain a Dirichlet distribution $\text{Dir}(\alpha^0(s,a), \alpha^1(s,a), \ldots, \alpha^N(s,a))$ as the prior for the Q-value node at state $s$, action $a$. At each time step $t$ we sample $L_t(s,a) \sim \text{Dir}(\alpha^0(s,a), \alpha^1(s,a), \ldots, \alpha^N(s,a))$ and compute $\overline{\phi}_t(s,a) = [z_0(s,a), z_1(s,a), \ldots, z_N(s,a)]^\top L_t(s,a)$. Then, the action $a_t$ is selected as follows:

$$a_t = \arg\max_a \left\{ \overline{\phi}_t(s,a) \right\}$$

After taking action $a_t$ and get an intermediate reward $\overline{Q}_t(s,a_t) = r_t(s,a_t) + \gamma \widehat{V}_t(s')$. The posterior is also a Dirichlet: $\text{Dir}(\alpha^0(s,a), \ldots, \alpha^t(s,a) + 1, \ldots, \alpha^N(s,a))$ with the intermediate reward at time step $t$: $\overline{Q}_t(s,a_t)$ is in the range of the atom $z_t(s,a)$. We denote this mechanism as Categorical Thompson sampling for Tree Search (CATS) method.

**Paricle based distribution**: In the particle-based approach, the prior Dirichlet distribution of the Q-value node at state $s$, action $a$ is $\text{Dir}(\alpha(s,a))$, with $\alpha(s,a)$ is initiated as [1]. Considering each Q value distribution at state $s$, action $a$ has a set of particle $\{\overline{Q}_t(s,a)\}$ with the corresponding weighted $\alpha(s,a) = \{\alpha^t(s,a)\}$ At each time step $t$ we also sample $L_t(s,a) \sim \text{Dir}(\alpha(s,a))$ and compute $\overline{\phi}_t(s,a) = [1, \overline{Q}_0(s,a), \overline{Q}_1(s,a), \ldots, \overline{Q}_N(s,a)]^\top L_t(s,a)$. Then the action $a_t$ is chosen as

$$a_t = \arg\max_a \left\{ \overline{\phi}_t(s,a) \right\}.$$

After taking action $a_t$ and get an intermediate reward $\overline{Q}_t(s,a_t) = r_t(s,a_t) + \gamma \widehat{V}_t(s')$. We update $\alpha^t(s,a) = \alpha^t(s,a) + 1$ if $\overline{Q}_t(s,a_t)$ is in the set $\{\overline{Q}_t(s,a)\}$. If not, we add $\overline{Q}_t(s,a_t)$ to the set $\{\overline{Q}_t(s,a)\}$ and add 1 to the set $\{\alpha^t(s,a)\} = \{\alpha^t(s,a), 1\}$.

We call this method as Paricle Thompson sampling for Tree Search (PATS) method. Detailed pseudocode and a comparison of CATS and PATS can be seen in Fig 1. The two methods are identical in all procedures except for the Q value function backup (**SimulateQ**) and the action selection function (**SelectAction**).

**Remark 1.** *CATS and PATS both use similar action selection strategies within a bandit setting, specifically referring to Multinomial Thompson Sampling and Non-Parametric Thompson Sampling, respectively (29). While CATS action selection heavily depends strictly on Thompson Sampling by maintaining parameters of posterior Q-value distribution, PATS is not based on the posterior sampling in the strict sense. At each step, it computes an average of the observed rewards with random weight and is a Non-Parametric approach. Furthermore, CATS maintains a fixed set of atoms, whereas in PATS, the number of particles increases depending on the observed Q values.*

In the next section, we provide a theoretical analysis of the convergence of simple regret for CATS and PATS.

| **Algorithm 3** CATS in Non-stationary bandits | **Algorithm 4** PATS in Non-stationary bandits |
|---|---|
| **Require:** K arms; n: number of plays; 
 $N + 1$ support size of categorical distributions 
 Init $(\alpha_a^0, \ldots, \alpha_a^N) = (1, \ldots, 1)$ for each $a \in [K]$ 
 **Main** *()* 
    **for** *t = 0,1,2,…, n* **do** 
      **for** $a \in [A]$ **do** 
        $L_{a,t} \sim \mathrm{Dir}(\alpha_a^0, \ldots, \alpha_a^N)$ 
        $\overline{\phi}_{a,t} = [0, \frac{R(t)}{N}, \frac{2R(t)}{N}, \cdots, R(t)]^\top L_t$ 
      $a = \arg\max_a \{\overline{\phi}_{a,t}\}$ 
      Pull arm $a$ and observe reward 
      $R_{a,t} = \frac{mR(t)}{N}$ where $m \in \{0, 1, \ldots N\}$ 
      Update $\alpha_a^{\hat{m}} = \alpha_a^m + 1$ | **Require:** K arms; n: number of plays; 
 Init $\alpha_a = (1)$; $\mathcal{S}_a = (1)$ for each $a \in [K]$ 
 **Main** *()* 
    **for** *t = 0,1,2,…, n* **do** 
      **for** $a \in [A]$ **do** 
        $L_{a,t} \sim \mathrm{Dir}(\alpha_a)$ 
        $\overline{\phi}_{a,t} = \mathcal{S}_a^\top L_{a,t}$ 
      $a = \arg\max_a \{\overline{\phi}_{a,t}\}$ 
      Pull arm $a$ and observe reward $R_{a,t}$ 
      **if** $R_{a,t} \in \{\mathcal{S}_a\}$ **then** 
        $\alpha_a^t \mathrel{+}= 1$ //$\alpha_a^t$ : weight of $R_{a,t}$ 
      **else** 
        $\mathcal{S}_a := (\mathcal{S}_a, R_{a,t})$ 
        $\alpha_a := (\alpha_a, 1)$ |

Figure 2: Comparing CATS (left) and PATS (right) in Non-stationary bandits.

## 4 Theoretical analysis

Planning in MCTS involves making a sequence of decisions along the tree, where each internal node functions as a non-stationary bandit, with the empirical mean drifting due to the action selection strategy. Therefore, we first study the non-stationary multi-armed bandit settings using the action selections of CATS and PATS, examining the concentration properties of the power mean backup for each arm relative to the optimal arm. We then apply these results to MCTS.

### 4.1 Non-stationary multi-armed bandit

We consider a class of non-stationary multi-armed bandit (MAB) problems with $K \geq 1$ arms. Let $R_{a,t}$ denote the random reward obtained by playing arm $a \in [K]$ at the time step $t$ bounded in $[0, R]$. We consider $\widehat{\mu}_{a,n} = \frac{1}{n} \sum_{t=1}^{n} R_{a,t}$ as the average rewards collected at arm $a$ after n plays. We first define:

**Definition 1.** *A sequence of estimators $(\widehat{V}_n)_{n \geq 1}$ is concentrated and convergent towards some limit $V$ if the following two properties hold:*

     *(A) Concentration: For all $n \geq 1$, for all $\varepsilon > 0$, $\exists c > 0$ that $\mathbb{P}\left(|\widehat{V}_n - V| > \varepsilon\right) \leq cn^{-1}\varepsilon^{-1}$.*

     *(B) Convergence: $\lim_{n \to \infty} \mathbb{E}[\widehat{V}_n] = V$.*

*In that case, we write $\operatorname{plim}_{n \to \infty} \widehat{V}_n = V$.*

We assume that the reward sequence $\{R_{a,t}\}, t \geq 1$ is a non-stationary process satisfying the convergence and concentration properties from Definition 1, by making the following assumption:

**Assumption 1.** *Consider K arms that for $a \in [K]$, let $(\widehat{\mu}_{a,n})_{n \geq 1}$ be a sequence of estimator satisfying*

$$\operatorname*{plim}_{n \to \infty} \widehat{\mu}_{a,n} = \mu_a.$$

The action selection of CATS and PATS follows closely as in Section 3.3 and pseudocode are shown in Fig. 2. Let us define $\widehat{\mu}_n(p) = \left(\sum_{a=1}^{K} \frac{T_a(n)}{n} \widehat{\mu}_{a,T_a(n)}^p\right)^{\frac{1}{p}}$ as the power mean value backup operator after $n$ rounds. Here $1 \leq p < \infty$ is a constant. We denote $T_a(n)$ is the number of visitations of the arm $a$.

We define $\mu_\star = \max_{a \in [K]}\{\mu_a\}$ and assume that $\mu_\star$ is unique. Then, we establish the concentration and convergence properties of the power mean backup operator $\widehat{\mu}_n(p)$ towards the optimal value $\mu_\star$, as shown in Theorem 1 and Theorem 2, respectively for CATS and PATS.

**Theorem 1.** *For $a \in [K]$, let $(\widehat{\mu}_{a,n})_{n \geq 1}$ be a sequence of estimator satisfying $\operatorname*{plim}_{n \to \infty} \widehat{\mu}_{a,n} = \mu_a$ and let $\mu_\star = \max_a\{\mu_a\}$. Assume that all the estimators are bounded in $[0, R]$. We consider a bandit algorithm that selects each arm according to CATS once in each round $n \geq K$. Then, $\operatorname*{plim}_{n \to \infty} \widehat{\mu}_n(p) = \mu_\star$.*

**Theorem 2.** *For $a \in [K]$, let $(\widehat{\mu}_{a,n})_{n \geq 1}$ be a sequence of estimator satisfying $\operatorname*{plim}_{n \to \infty} \widehat{\mu}_{a,n} = \mu_a$ and let $\mu_\star = \max_a\{\mu_a\}$. Assume that all the estimators are bounded in $[0, R]$. We consider a bandit algorithm that selects each arm according to PATS once in each round $n \geq K$. Then, $\operatorname*{plim}_{n \to \infty} \widehat{\mu}_n(p) = \mu_\star$.*

Detailed proofs of the two Theorems can be found in the appendix. Based upon these results we analyse the concentration properties for any internal node and convergence of the simple regret in the MCTS in the next section.

## 4.2 Monte-Carlo Tree Search

Before presenting the main results (Theorem 3 Theorem 4), we first show an important Lemma

**Lemma 1.** *Let $(\widehat{V}_{m,n})_{n \geq 1}$, $m \in [M]$, be a sequence of estimator satisfying $\underset{n \to \infty}{plim} \widehat{V}_{m,n} = V_m$.*

*Assume that there exists a constant $L > 0$ such that $L = supremum\{\widehat{V}_{m,n}\}_{n \geq 1}$. Let $R_i$ be an iid sequence with mean $\mu$ and $S_i$ be an iid sequence from a distribution $p = (p_1, \ldots, p_M)$ supported on $\{1, \ldots, M\}$. Introducing the random variables $N_m^n = \#|\{i \leq n : S_i = s_m\}|$, we define the sequence of estimator*

$$\widehat{Q}_n = \frac{1}{n} \sum_{i=1}^n R_i + \gamma \sum_{m=1}^M \frac{N_m^n}{n} \widehat{V}_{m,N_m^n}.$$

*Then $\underset{n \to \infty}{plim} \widehat{Q}_n = \mu + \sum_{m=1}^M p_m V_m$.*

The significance of Lemma 1 lies in demonstrating the concentration and convergence of an estimated Q value, conditioned on the concentration and convergence of a child V-value node. Here, $\widehat{V}_{\cdot,n}$ represents the value estimation at time step $n$, and $R_i$ denotes an intermediate reward received by taking a specific action at a particular state.

Next, we first start with Theorem 3 to show the convergence and concentration of any V-Node and Q-node in the tree for CATS.

**Theorem 3.** *When we apply the CATS algorithm, we have*

    *(i) For any node $s_h$ at the depth $h^{th}$ in the tree, $\underset{n \to \infty}{plim} \widehat{Q}_n(s_h, a_k) = \widetilde{Q}(s_h, a_k)$.*

    *(ii) For any node $s_h$ at the depth $h^{th}$ in the tree, $\underset{n \to \infty}{plim} \widehat{V}_n(s_h) = \widetilde{V}(s_h)$.*

We can derive a similar result for PATS as shown in Theorem 4.

**Theorem 4.** *When we apply the PATS algorithm, we have*

    *(i) For any node $s_h$ at the depth $h^{th}$ in the tree, $\underset{n \to \infty}{plim} \widehat{Q}_n(s_h, a_k) = \widetilde{Q}(s_h, a_k)$.*

    *(ii) For any node $s_h$ at the depth $h^{th}$ in the tree, $\underset{n \to \infty}{plim} \widehat{V}_n(s_h) = \widetilde{V}(s_h)$.*

The results of Theorems 4 and 4 demonstrate that, at any node in the tree, both the V-value and Q-value nodes are convergent and concentrated. These results are applicable to any power mean backup operator of V-value nodes with $p \in [1, +\infty)$. Finally, we show important results in Theorem 5, and Theorem 6, since they show the convergence of simple regret of CATS and PATS, respectively.

**Theorem 5.** *(Convergence of Simple Regret of CATS) We have at the root node $s_0$,*

$$\left| \mathbb{E}\left[ V^\star(s_0) - \widehat{V}_n(s_0) \right] \right| \leq O(n^{-1}).$$

**Theorem 6.** *(Convergence of Simple Regret of PATS) We have at the root node $s_0$,*

$$\left| \mathbb{E}\left[ V^\star(s_0) - \widehat{V}_n(s_0) \right] \right| \leq O(n^{-1}).$$

**Remark 2.** *These results demonstrate that both CATS and PATS share the same convergence rate for value estimation at the root node of $\mathcal{O}(n^{-1})$, which improves over the rate $\mathcal{O}(n^{-1/2})$ of Fixed-Depth-MCTS (33). Furthermore, Our finding more broadly applies to the power mean estimator with $p \in [1, +\infty)$.*

# 5 Experiments

We compare our methods with UCT (21), Fixed-Depth-MCTS (33), MENTS (40), RENTS, TENTS (14), BTS (27) and DNG (1) in a stochastic setting (*SyntheticTree*) to highlight the benefits of CATS and PATS in stochastic environments. Additionally, we test on 17 Atari games, comparing our algorithms with DQN (base network without planning) and other non-distributional planning methods (Power-UCT (12), MENTS (40), TENTS (14)) to demonstrate CATS and PATS' competitiveness and put results in Appendix. In all settings, we use 100 atoms for CATS, and set the discount factor $\gamma$ to 0.99 for Atari, and $\gamma$ to 1 for *SyntheticTree*.

**SyntheticTree**: We evaluate CATS and PATS using the synthetic tree toy problem (14). This problem involves a tree with depth $d$ and branching factor $k$. Each tree edge has a random value between 0 and 1. Returns at the leaf nodes are simulated using Gaussian distributions with means equal to the sum of edge values from the root to the leaf, and a standard deviation of $0.5$. Means are normalized between 0 and 1. An agent traverses the tree from the root, aiming to find the leaf node with the highest mean value. Internal nodes give zero reward, while leaf nodes provide a reward sampled from their Gaussian distribution. We introduce stochasticity into the environment by altering the transition probabilities: there is a $50\%$ chance of moving to the intended node and a $50\%$ chance of moving to a different node with equal probability. We conduct 25 experiments on five trees with five runs each, covering all combinations of branching factors $k = \{2, 4, 6, 8, 10, 12, 14, 16, 100, 200\}$ and depths $d = \{1, 2, 3, 4\}$. We compute the value estimation error at the root node. Fig. 3 shows the convergence of the value estimations of CATS and PATS at the root node in the Synthetic Tree environment which shows they archives faster convergence compared to other methods.

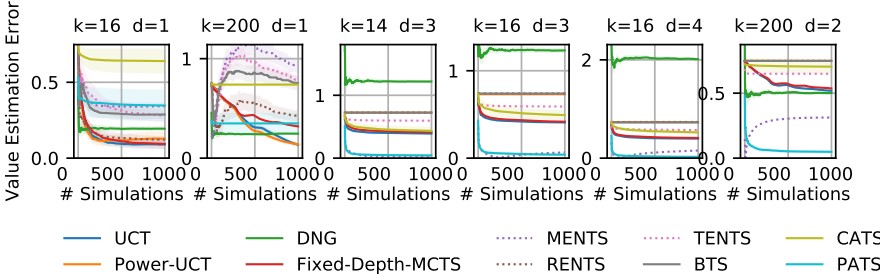

Figure 3: Performance of CATS and PATS in SyntheticTree.

# 6 Conclusion

To conclude, our work introduces Categorical Thompson Sampling for MCTS (CATS) and Particle Thompson Sampling for MCTS (PATS), distributional planning approaches specifically designed to tackle complexities arising from stochasticity. CATS uses a categorical distribution, while PATS uses a particle-based distribution to represent and model the uncertainty inherent in return outcomes. We also propose exploration strategies based on Thompson Sampling that leverage this distributional modeling. Our methods come with a rigorous theoretical convergence guarantee, achieving a simple regret polynomial decay of the order $O(n^{-1})$, which improves over the $O(n^{-1/2})$ rate of the fixed version of UCT (32). Empirical findings conclusively demonstrate the effectiveness of our approach in stochastic environments.

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

 # A  Outline

# B  Notations

Table 1: List of all notations for Non-stationary Multi-arms bandit.

| Notation | Type | Description |
|---|---|---|
| $K$ | $\mathbb{N}$ | Number of arms |
| $T_a(t)$ | $\mathbb{N}$ | Number of visitations at arm $a$ after $t$ timesteps |
| $\mu_a$ | $\mathbb{R}$ | mean value of arm $a$ |
| $a_\star$ | $\mathcal{A}$ | optimal action |
| $\mu_\star$ | $\mathbb{R}$ | mean value of an optimal arm. We assume it is unique. |
| $\widehat{\mu}_n(p)$ | $\mathbb{R}$ | power mean estimator, with a constant $p \in [1, +\infty)$ |
| $\widehat{\mu}_{a,n}$ | $\mathbb{R}$ | mean estimator of arm $a$ after $n$ visitations |

# C  Supporting Lemmas

We start with a result of the following lemma which plays an important role in the analysis of our MCTS algorithm.

**Lemma 1.** *For $m \in [M]$, let $(\widehat{V}_{m,n})_{n\geq 1}$ be a sequence of estimator satisfying $\underset{n\to\infty}{plim}\,\widehat{V}_{m,n} = V_m$. Assume that there exists a constant $L > 0$ such that $L = supremum\{\widehat{V}_{m,n}\}_{n\geq 1}$. Let $R_i$ be an iid sequence with mean $\mu$ and $S_i$ be an iid sequence from a distribution $p = (p_1, \ldots, p_M)$ supported on $\{1, \ldots, M\}$. Introducing the random variables $N_m^n = \#|\{i \leq n : S_i = s_m\}|$, we define the sequence of estimator*

$$\widehat{Q}_n = \frac{1}{n}\sum_{i=1}^{n} R_i + \gamma \sum_{m=1}^{M} \frac{N_m^n}{n} \widehat{V}_{m,N_m^n}.$$

*Then there exists some constant $c'$ (which depends on $p_i$ (i=1,2,...,M), $\gamma$, $\mu$) such that*

$$\underset{n\to\infty}{plim}\,\widehat{Q}_n = \mu + \sum_{m=1}^{M} p_m V_m.$$

*Proof.* Let $p = (p_1, p_2, ...p_M), p \in \triangle^M$ where $\triangle^M = \{x \in \mathbb{R}^M : \sum_{i=1}^{M} R_i = 1, R_i \geq 0\}$ is the $(M-1)$-dimensional simplex. Let us study a random vector $\widehat{p}_n = (\frac{N_1^n}{n}, \frac{N_2^n}{n}, ..., \frac{N_M^n}{n})$. Let us define

Table 2: List of all notations for Monte-Carlo Tree Search.

| Notation | Type | Description |
|---|---|---|
| $\gamma$ | $\mathbb{R}$ | Discount factor |
| N | $\mathbb{N}$ | Number of atoms |
| $s_h$ | $\mathcal{S}$ | state at depth $h$ |
| $\widehat{V}_t(s)$ | $\mathbb{R}$ | Estimated Value function at state $s$ after $t$ visitations |
| $T_s(t)$ | $\mathbb{N}$ | Number of visitations at state $s$ after $t$ timesteps |
| $T_{s,a}(t)$ | $\mathbb{N}$ | Number of visitations at $(s,a)$ after $t$ timesteps |
| $T_{s,a}^{s'}(t)$ | $\mathbb{N}$ | Number of visitations at $(s,a)$ that goes to $s'$ after $t$ timesteps |
| $\widehat{Q}_t(s,a)$ | $\mathbb{R}$ | Estimated Q Value function at state $s$ action $a$ after $t$ visitations |
| $Q_{\min}(s,a)$ | $\mathbb{R}$ | Minimum value for the Q value distribution at state $s$, action $a$ |
| $Q_{\max}(s,a)$ | $\mathbb{R}$ | Maximum value for the Q value distribution at state $s$, action $a$ |
| $\mathcal{R}(s,a)$ | | Reward distribution at state $s$ action $a$ |
| $\mathcal{V}(s)$ | | Value distribution at state $s$ |
| $\mathcal{Q}(s,a)$ | | Q Value distribution at state $s$ action $a$ |
| $p_i(s,a)$ | $\mathbb{R}$ | Probability of the $i_{th}$ atom at the Q Value distribution at state $s$ action $a$ |
| $\triangle z$ | $\mathbb{R}$ | Size of each atom |
| $z_i(s,a)$ | $\mathbb{R}$ | value of the atom $i^{th}$ at state $s$, action $a$. |
| $\overline{Q}_t(s,a)$ | $\mathbb{R}$ | intermediate Q value at time $t$ at $(s,a)$ |

$V = (V_1, V_2, ... V_M)$. Let $\widehat{R}_n = \frac{1}{n} \sum_{i=1}^{n} R_i$, $\widehat{V}_n = (\widehat{V}_{1,N_1^n}, \widehat{V}_{2,N_2^n}, ..., \widehat{V}_{M,N_M^n})$, $\sum_{i=1}^{M} N_i^n = n$, $N_i^n$ is the number of times that population $i$ was observed. We have $\widehat{Q}_n = \widehat{R}_n + \gamma \left\langle \widehat{p}_n, \widehat{V}_n \right\rangle$. Therefore,

$$\mathbb{P}\left( \widehat{Q}_n - (\mu + \gamma \langle p, V \rangle) \geq \epsilon \right) \leq \mathbb{P}\left( \widehat{R}_n - \mu \geq \frac{1}{2}\epsilon \right) + \mathbb{P}\left( \gamma \left\langle \widehat{p}_n, \widehat{V}_n \right\rangle - \gamma \langle p, Y \rangle \geq \frac{1}{2}\epsilon \right)$$

$$\leq \exp\{-2n\frac{\epsilon^2}{4}\} + \underbrace{\mathbb{P}\left( \left\langle \widehat{p}_n, \widehat{V}_n \right\rangle - \langle p, Y \rangle \geq \frac{1}{2\gamma}\epsilon \right)}_{A}.$$

To upper bound A, let us consider $\left\langle \widehat{p}_n, \widehat{V} \right\rangle - \langle p, V \rangle = \left\langle (\widehat{p}_n - p), \widehat{V}_n \right\rangle + \left\langle p, (\widehat{V} - V) \right\rangle$. Then,

$$A \leq \underbrace{\mathbb{P}\left( \left\langle (\widehat{p}_n - p), \widehat{V}_n \right\rangle \geq \frac{1}{4\gamma}\epsilon \right)}_{A_1} + \underbrace{\mathbb{P}\left( \left\langle p, (\widehat{V}_n - V) \right\rangle \geq \frac{1}{4\gamma}\epsilon \right)}_{A_2}.$$

488  By applying a Hölder inequality to $\widehat{p}_n - p$ and $\widehat{V}$, we obtain

$$\left\langle (\widehat{p}_n - p), \widehat{V}_n \right\rangle \leq \parallel \widehat{p}_n - p \parallel_1 \parallel \widehat{V}_n \parallel_\infty = \parallel \widehat{p}_n - p \parallel_1 L,$$

489  with $L$ is the supremum of $\widehat{V}$. Then we can derive

$$A_1 = \mathbb{P}\left( \left\langle (\widehat{p}_n - p), \widehat{V}_n \right\rangle \geq \frac{1}{4\gamma}\epsilon \right) \leq \mathbb{P}\left( \parallel \widehat{p}_n - p \parallel_1 L \geq \frac{1}{4\gamma}\epsilon \right)$$
$$= \mathbb{P}\left( \parallel \widehat{p}_n - p \parallel_1 \geq \frac{1}{4\gamma L}\epsilon \right).$$

490  According to (39), we have for any $M \geq 2$ and $\delta \in [0, 1]$

$$\mathbb{P}\left( \parallel \widehat{p}_n - p \parallel_1 \geq \sqrt{\frac{2M \ln(2/\delta)}{n}} \right) \leq \delta.$$

491  Define $\epsilon = \sqrt{\frac{2M \ln(2/\delta)}{n}}$, therefore $\delta = 2 \exp\{\frac{-n\epsilon^2}{2M}\}$, we have

$$\mathbb{P}\left( \parallel \widehat{p}_n - p \parallel_1 \geq \epsilon \right) \leq 2 \exp\{\frac{-n\epsilon^2}{2M}\}.$$

492  Therefore,

$$A_1 \leq \mathbb{P}\left( \parallel \widehat{p}_n - p \parallel_1 \geq \epsilon \right) \leq 2 \exp\{\frac{-n\epsilon^2}{32M\gamma^2 L^2}\}.$$

493  We also have

$$A_2 = \mathbb{P}\left( \sum_{m=1}^{M} p_m(\widehat{V}_{m,N_m^n} - V_m) \geq \frac{1}{4\gamma}\epsilon \right)$$
$$\leq \sum_{m=1}^{M} \mathbb{E}\left[ \mathbb{P}\left( \frac{1}{N_m^n} \sum_{t=1}^{N_m^n} V_{m,t} - V_m \geq \frac{1}{4\gamma p_m}\epsilon \Big| N_m^n \right) \right]$$
$$\leq \sum_{m=1}^{M} \mathbb{E}\left[ c(N_m^n)^{-1}(\frac{\epsilon}{4\gamma p_m})^{-1} \right].$$

494  Let us define an event $\mathcal{E} = \left\{ N_m^n \geq \frac{np_m}{2} \right\}$. Therefore,

$$A_2 \leq \sum_{m=1}^{M} \mathbb{E}\left[ c(\frac{np_m}{2})^{-1}(\frac{\epsilon}{4\gamma p_m})^{-1} \right]$$
$$+ \sum_{m=1}^{M} \mathbb{E}\left[ \mathbb{P}(N_m^n < \frac{np_m}{2}) \right] = \sum_{m=1}^{M} (c2^{1+2}\gamma^1 p_m^{-1+1})n^{-1}\epsilon^{-1}$$
$$+ \sum_{m=1}^{M} \mathbb{E}\left[ \mathbb{P}(N_m^n - p_m n \leq -\frac{p_m n}{2}) \right]$$
$$\leq \sum_{m=1}^{M} (c2^3\gamma)n^{-1}\epsilon^{-1} + \sum_{m=1}^{M} \exp\left\{ -2n(\frac{p_m n}{2})^2 \right\}$$

495  We consider $p_m > 0$ only since if $p_m = 0, p_m(\widehat{V}_{m,N_m^n} - V_m) = 0$, and has been eliminated.
496  Therefore,

$$A \leq A_1 + A_2 \leq 2 \exp\{\frac{-n\epsilon^2}{32M\gamma^2 L^2}\} + \sum_{m=1}^{M} (c2^3\gamma)n^{-1}\epsilon^{-1} + \sum_{m=1}^{M} \exp\left\{ -2n(\frac{p_m n}{2})^2 \right\}.$$

That leads to

$$\mathbb{P}\left(\widehat{Q}_n - \left(\mu + \gamma \langle p, V \rangle\right) \geq \epsilon\right) \leq \exp\{-2n\frac{\epsilon^2}{4}\}$$

$$+ 2\exp\{\frac{-n\epsilon^2}{32M\gamma^2 L^2}\} + \sum_{m=1}^{M}(c2^3\gamma)n^{-1}\epsilon^{-1} + \sum_{m=1}^{M}\exp\left\{-2n(\frac{p_m n}{2})^2\right\} \leq c'n^{-1}\epsilon^{-1},$$

with $c' > 0$ depends on $c, M, p_i$. So that

$$\mathbb{P}\left(\widehat{Q}_n - \left(\mu + \gamma \langle p, V \rangle\right) \geq \epsilon\right) \leq c'n^{-1}\epsilon^{-1},$$

By following the same steps, we can derive

$$\mathbb{P}\left(\widehat{Q}_n - \left(\mu + \gamma \langle p, V \rangle\right) \leq -\epsilon\right) \leq c'n^{-1}\epsilon^{-1}.$$

Therefore, with $n \geq 1, \epsilon > 0$,

$$\mathbb{P}\left(\left|\widehat{Q}_n - \left(\mu + \gamma \langle p, V \rangle\right)\right| \geq \epsilon\right) \leq c'n^{-1}\epsilon^{-1}.$$

Furthermore,

$$\widehat{Q}_n - \left(\mu + \gamma \langle p, V \rangle\right) = (\widehat{R}_n - \mu) + \left(\gamma \left\langle \widehat{p}_n, \widehat{V}_n \right\rangle - \gamma \langle p, Y \rangle\right)$$

$$= (\widehat{R}_n - \mu) + \gamma\left(\left\langle (\widehat{p}_n - p), \widehat{V}_n \right\rangle + \left\langle p, (\widehat{V} - V) \right\rangle\right)$$

Therefore,

$$\Rightarrow \left|\mathbb{E}[\widehat{Q}_n] - \left(\mu + \gamma \langle p, V \rangle\right)\right| \leq \left|\mathbb{E}[(\widehat{R}_n - \mu)]\right| + \gamma\left(\left|\mathbb{E}[\widehat{p}_n - p]\right|\left|\widehat{V}_n\right| + p\left|\mathbb{E}[\widehat{V} - V]\right|\right)$$

$$\Rightarrow \left|\mathbb{E}[\widehat{Q}_n] - \left(\mu + \gamma \langle p, V \rangle\right)\right| \leq \left|\mathbb{E}[(\widehat{R}_n - \mu)]\right| + \gamma\left(L\left|\mathbb{E}[\widehat{p}_n - p]\right| + p\left|\mathbb{E}[\widehat{V} - V]\right|\right)$$

Also because $\lim_{n\to\infty}\mathbb{E}[\widehat{V}_{m,n}] = V_m$, $\lim_{n\to\infty}\frac{\widehat{N}_m^n}{n} = p_m$, and $\mathbb{E}[(\widehat{R}_n - \mu)] = 0$ so that,

$$\lim_{n\to\infty}\mathbb{E}[\widehat{Q}_n] = \mu + \gamma\sum_{m=1}^{M}p_m V_m.$$

That mean

$$\plim_{n\to\infty}\widehat{Q}_n = \mu + \gamma\sum_{m=1}^{M}p_m V_m,$$

which concludes the proof. $\square$

Results from Lemma 1 is important as it shows the concentration for the Q value estimation given the concentration of V value of the children nodes.

**Lemma 2.** *Let consider non-negative variables $x, y \in \mathbb{R}^+$, and a constant m that $0 \leq m \leq 1$. Then*

$$(x + y)^m \leq x^m + y^m.$$

*Proof.* With $y = 0$, or $x = 0$, the inequality (2) becomes correct. Let consider the case where $x > 0, y > 0$, the inequality (2) can be written as

$$(\frac{x}{y} + 1)^m \leq \left(\frac{x}{y}\right)^m + 1.$$

Let us define a function

$$f(t) = (t + 1)^m - t^m - 1, (t > 0).$$

512 We can see that
$$f^{'}(t) = m(t+1)^{m-1} - mt^{m-1} = m\left((t+1)^{m-1} - t^{m-1}\right) \le 0 \text{ with } m \in [0,1], t > 0,$$

513 because $g(x) = x^{m-1}$ is a decreasing function with $m \in [0,1], x > 0$. Therefore,
$$f(t) \le f(0) = 0 \text{ with } t > 0.$$

514 So that,
$$(t+1)^m - t^m - 1 \le 0, (t > 0).$$

515 with $t = \frac{x}{y} \ge 0$, we can derive the inequality (2). □

516 We use Minkowski's inequality as shown below

517 **Lemma 3.** *(Minkowski's inequality) Given $p \ge 1, \{x_i, y_i\} \in \mathbb{R}, i = 1, 2, ..., n$, then we have the*
518 *following inequality*

$$\left(\sum_i (|x_i + y_i|)^p\right)^{\frac{1}{p}} \le \left(\sum_i (|x_i|)^p\right)^{\frac{1}{p}} + \left(\sum_i (|y_i|)^p\right)^{\frac{1}{p}}.$$

519 *Proof.* This is a basic result. □

520 **Lemma 4.** *(Markov's inequality) If $X$ is a nonnegative random variable and $a > 0$, then the*
521 *probability that $X$ is at least $a$ is at most the expectation of X divided by a:*

$$\mathbf{Pr}(X > a) \le \frac{\mathbb{E}[X]}{a}.$$

522 *Proof.* This is a well-known result. □

## D  Convergence of CATS and PATS in Non-stationary multi-armed bandits

524 We note that in an MCTS tree, each node is considered a non-stationary multi-armed bandit where
525 the average mean drifts due to the given action selection strategy. Therefore, we first study the
526 convergence of CATS and PATS in non-stationary multi-armed bandits where the action selection is
527 Thompson sampling, with the power mean backup operator at the root node. Detailed descriptions of
528 the CATS and PATS in Non-stationary multi-armed bandits settings can be found in the main article
529 in the Theoretical Analysis section.

530 We first establish the convergence and concentration properties for the power mean backup operator
531 in non-stationary bandits, detailed in Theorem 1 for CATS and Theorem 2 for PATS.

532 To achieve these results, we demonstrate that the expected payoff of the power mean backup operator
533 decays polynomially at a rate of $O(\frac{\log n}{n})$. This is supported by Lemma 7 for CATS and Lemma 8 for
534 PATS. Critical to this analysis are Lemma 5 and Lemma 6, which establish an upper bound of $\log(n)$
535 for the expected number of suboptimal arm pulls.

536 We introduce some important definitions. $F_a^n$ represents the empirical cumulative distribution function
537 of arm $a$ after $n$ visitations, and $F_a$ represents the cumulative distribution function of arm $a$. We
538 employ the following distance measure: If $P$ and $Q$ are two distributions characterized by parameters
539 $p = (p_0, p_1, \cdots, p_N)$ and $q = (q_0, q_1, \cdots, q_N)$ respectively, then the distance is defined as

$$d(P, Q) := \parallel p - q \parallel_\infty = \sup_{i \in [0,N]} |p_i - q_i|$$

540 This represents the $L^\infty$ distance between $p$ and $q$ in $\mathbb{R}^{N+1}$. We also denotes
541 $\text{KL}(P \parallel Q)$ as the Kullback–Leibler divergence between $P$ and $Q$, and denote
542 $\mathcal{K}_{\text{inf}}(F_a, \mu_\star) = \inf_{G:\mathbb{E}[G]>\mu_\star} \text{KL}(F_a \parallel G)$. In addition, we denote $\mathcal{K}_{\text{inf}}^{(N)}(F_a, \mu_\star) =$
543 $\inf\left\{\text{KL}(F_a \parallel G) \middle| \text{the support of G} \in \left\{0, \frac{R}{N}, \frac{2R}{N}, \cdots, R\right\}, \mathbb{E}[G] > \mu_\star\right\}$.

544 We see that the definition of $\mathcal{K}_{\text{inf}}(F_a, \mu_\star)$ and $\mathcal{K}_{\text{inf}}^{(N)}(F_a, \mu_\star)$ is only difference in the support set.

We denote the true parameter of arm $a$ by $p_a = (p_a^0, p_a^1, \ldots, p_a^N)$ with $p_a^i = \mathbf{Pr}_{X \sim F_a}[X = \frac{i}{N}]$. We denote the parameter of the posterior distribution of arm $a$ as $\alpha_a = (\alpha_a^0, \alpha_a^1, \ldots, \alpha_a^N)$. Since each arm $a$ is non-stationary, we also denote the parameter of arm $a$ after $n$ visitations by $p_a(n) = (p_a^0(n), p_a^1(n), \ldots, p_a^N(n))$ with $p_a^i(n) = \mathbf{Pr}_{X \sim F_a^n}[X = \frac{i}{N}]$. The parameter of the posterior distribution of arm $a$ denoted as $\alpha_a(n) = (\alpha_a^0(n), \alpha_a^1(n), \ldots, \alpha_a^N(n))$ We first show the results of an important Lemma 5. The proof follows closely to the Proof of Proposition 7 (29). The only difference is that in our settings, we study non-stationary bandits.

**Lemma 5.** *Consider Categorical Thompson Sampling(CATS) strategy applied to a non-stationary problem where the pay-off sequence satisfies Assumption 1. Let $T_a(n)$ denote the number of plays of arm $a$ up to timestep $n$.*

*If $a$ is the index of a suboptimal arm, Then for any $\epsilon_0, \epsilon_1 \geq 0$, each sub-optimal arm $a$ is played in expectation at most*

$$\mathbb{E}[T_a(n)] \leq \frac{(1 + \epsilon_0) \log n}{\mathcal{K}_{inf}^{(N)}(F_a, \mu_\star) - \epsilon_1} + o(\log n) + O(1),$$

*Proof.* We have $\overline{\phi}_{a,t} = [0, \frac{R}{N}, \frac{2R}{N}, \cdots, R]^\top L_{a,t}$, with $L_{a,t} \sim \text{Dir}(\alpha_a^0(t), \ldots, \alpha_a^N(t))$.

To analyze the expectation associated with selecting a suboptimal arm $a$, we decompose it into two components:

$$\mathbb{E}\left[\sum_{t=1}^n \mathbb{1}(I(t) = a)\right] = \underbrace{\mathbb{E}\left[\sum_{t=1}^n \mathbb{1}(I(t) = a), \overline{\phi}_{a,t} \geq \mu_* - \epsilon_1, d(\widehat{F}_{I(t)}, F_{I(t)}) \leq \epsilon_2)\right]}_{A1}$$

$$+ \underbrace{\mathbb{E}\left[\sum_{t=1}^n \mathbb{1}(I(t) = a), \overline{\phi}_{a,t} < \mu_* - \epsilon_1, d(\widehat{F}_{I(t)}, F_{I(t)}) > \epsilon_2)\right]}_{A2}$$

We first find an upper bound for $A_1$:

$$A1 = \sum_{t=1}^n \sum_{m=1}^n \mathbb{1}\left(I(t) = a, \overline{\theta}_k(t) \geq \mu_\star - \epsilon_1; \| \frac{\alpha_a(t)}{T_k(t) + N + 1} - p_a(t) \|_\infty \leq \epsilon_2, T_k(t) = m\right)$$

We see that if the event

$$\left\{I(t) = a, \overline{\theta}_k(t) \geq \mu_\star - \epsilon_1; \| \frac{\alpha_a(t)}{T_k(t) + N + 1} - p_a(t) \|_\infty \leq \epsilon_2, T_k(t) = m\right\}$$

occurs at step t for a certain $m \in [1, n]$, then $T_k(t') > T_k(t) = m$ for any $t' > t$. Therefore, for any $m \in [n]$

$$\sum_{t=1}^n \mathbb{1}\left(I(t) = a, \overline{\theta}_k(t) \geq \mu_\star - \epsilon_1; \| \frac{\alpha_a(t)}{T_k(t) + N + 1} - p_a(t) \|_\infty \leq \epsilon_2, T_k(t) = m\right) \leq 1$$

We can bound for any $m_0 \in [n]$

$$A1 \leq m_0 + \sum_{t=1}^n \sum_{m=m_0}^n \mathbb{E}\left[\mathbb{1}\left(I(t) = a, \overline{\theta}_k(t) \geq \mu_\star - \epsilon_1; \| \frac{\alpha_a(t)}{T_k(t) + N + 1} - p_a(t) \|_\infty \leq \epsilon_2, T_k(t) = m\right)\right]$$

$$\leq m_0 + \sum_{t=1}^n \sum_{m=m_0}^n \mathbf{Pr}\left(\overline{\theta}_k(t) \geq \mu_\star - \epsilon_1; \| \frac{\alpha_a(t)}{T_k(t) + N + 1} - p_a(t) \|_\infty \leq \epsilon_2, T_k(t) = m\right)$$

$$\leq m_0 + \sum_{t=1}^n \sum_{m=m_0}^n \mathbf{Pr}\left(\overline{\theta}_k(t) \geq \mu_\star - \epsilon_1 \middle| \| \frac{\alpha_a(t)}{T_k(t) + N + 1} - p_a(t) \|_\infty \leq \epsilon_2, T_k(t) = m\right)$$

$$\times \mathbf{Pr}\left(\| \frac{\alpha_a(t)}{T_k(t) + N + 1} - p_a(t) \|_\infty \leq \epsilon_2, T_k(t) = m\right) \tag{4}$$

By applying results of Lemma 13 Appendix F (29), we have

$$\mathbf{Pr}\left(\bar{\theta}_k(t) \geq \mu_\star - \epsilon_1 \middle| \alpha_a, T_k(t) = m\right)$$

$$\leq C(m + N + 1)^{N/2} \exp\{-(m + N + 1)\mathrm{KL}(P_{\alpha_a(t)} \parallel P^*_{\mu_\star - \epsilon_1})\}$$

where $P^*_{\mu_\star - \epsilon_1} = \arg\min_{x:u^\top x \geq \mu_\star - \epsilon_1} \mathrm{KL}(P_{\alpha_a} \parallel x)$ and $P_{\alpha_a(t)} = \frac{1}{n+N+1}\alpha_a(t)$. And by definition $\mathrm{KL}(P_{\alpha_a(t)} \parallel P^*_{\mu_\star - \epsilon_1}) = \mathcal{K}_{\inf}(P_{\alpha_a(t)}, \mu_\star - \epsilon_1)$, therefore

$$\mathbf{Pr}\left(\bar{\theta}_k(t) \geq \mu_\star - \epsilon_1 \middle| \alpha_a(t), T_k(t) = m\right)$$

$$\leq C(m + N + 1)^{N/2} \exp\{-(m + N + 1)\mathcal{K}_{\inf}(P_{\alpha_a(t)}, \mu_\star - \epsilon_1)\},$$

where $C = \frac{\exp\{1/12\}}{\Gamma(N+1)}\left(\frac{1}{\sqrt{2\pi}}\right)^N$. On the other hand, $\mathcal{K}_{\inf}(x, \mu_\star - \epsilon_1)$ is continuous in $x \in [0, 1]^{N+1}$ on the probability simplex with respect to the $L^\infty$ distance from ((19), Theorem 7) and Lemma 18 in Appendix H (29). Therefore, for any $\epsilon_3 > 0$, there exists $\epsilon_2 > 0$ and constant $C' > 0$ such that

$$\mathbf{Pr}\left(\bar{\theta}_k(t) \geq \mu_\star - \epsilon_1 \middle| \parallel \frac{\alpha_a(t)}{T_k(t) + N + 1} - p_a(t) \parallel_\infty \leq \epsilon_2, T_k(t) = m\right)$$

$$\leq C' \exp\{-(m + N + 1)(\mathcal{K}_{\inf}(p_a, \mu_\star - \epsilon_1) - \epsilon_3)\}$$

And because $\mathbf{Pr}\left(\parallel \frac{\alpha_a(t)}{T_k(t) + N + 1} - p_a(t) \parallel_\infty \leq \epsilon_2, T_k(t) = m\right) \leq 1$. Therefore,

$$A1 \leq m_0 + C'_1 \sum_{t=1}^{n} \exp\{-(m + N + 1)(\mathcal{K}_{\inf}(p_a, \mu_\star - \epsilon_1) - \epsilon_3)\}$$

$$\leq m_0 + C'_1 T \exp\{-(m + N + 1)(\mathcal{K}_{\inf}(p_a, \mu_\star - \epsilon_1) - \epsilon_3)\} \tag{5}$$

Choosing $m_0 = \frac{\log n}{\mathcal{K}_{\inf}(p_a, \mu_\star - \epsilon_1) - \epsilon_3} - N - 1$, we have

$$A1 \leq \frac{\log n}{\mathcal{K}_{\inf}(p_a, \mu_\star - \epsilon_1) - \epsilon_3} - N - 1 + C'_1$$

Furthermore, as from ((19), Theorem 7), it is proven that $\mu \to \mathcal{K}_{\inf}(F, \mu)$ is continuous for $\mu < 1$, when we scale reward from [0,1] to $[0, R]$ therefore $\mu$ from [0,1] to $[0, R]$. We have $\mu \to \mathcal{K}_{\inf}(F, \mu)$ is continuous for $\mu < R$. Therefore, $\forall \epsilon_4 > 0, \exists \epsilon_1 > 0$, such that

$$|\mathcal{K}_{\inf}(p_a, \mu^* - \epsilon_1) - \mathcal{K}_{\inf}(p_a, \mu^*)| \leq \epsilon_4$$
$$\Rightarrow \mathcal{K}_{\inf}(p_a, \mu^* - \epsilon_1) - \epsilon_3 \geq \mathcal{K}_{\inf}(p_a, \mu^*) - \epsilon_3 - \epsilon_4$$

Therefore, $\forall \epsilon_0 > 0$

$$A1 \leq \frac{(\epsilon_0 + 1)\log n}{\mathcal{K}_{\inf}(p_a, \mu_\star)} - N - 1 + C'_1$$

Also According to Proposition 8 (29), for any $\epsilon_0 > 0$ we have

$$A2 \leq O(1) \tag{6}$$

Combining inequality (5) and inequality (6) leads us to

$$\mathbb{E}[T_a(n)] \leq \frac{(1 + \epsilon_0)\log n}{\mathcal{K}_{\inf}^{(N)}(F_a, \mu_\star)} + o(\log n) + O(1).$$

Therefore which concludes the proof. $\qquad\square$

**Lemma 6.** *Consider Particle Thompson Sampling(PATS) strategy applied to a non-stationary problem where the pay-off sequence satisfies Assumption 1. Then for any $\epsilon_0 \geq 0$. Let $T_a(n)$ denote the number of plays of arm $a$ up to timestep $n$. Then if $a$ is the index of a suboptimal arm, then each sub-optimal arm $a$ is played in expectation at most*

$$\mathbb{E}[T_a(n)] \leq \frac{\log n}{\mathcal{K}_{inf}(F_a, \mu_\star) - \epsilon_0} + o(\log n) + O(1).$$

*Proof.* In this Theorem, we use the Levy distance. Recall that the Levy distance between two cumulative distribution functions $F$ and $G$ on $[0, 1]$ is defined as

$$D_L(F, G) = \inf\{\epsilon > 0 : \forall x \in [0, 1], F(x - \epsilon) - \epsilon \leq G(x) \leq F(x + \epsilon) + \epsilon\}.$$

The proof follows the same steps as in Lemma 5. We also can derive

$$\mathbb{E}\left[\sum_{t=1}^{n} \mathbb{1}(I(t) = a)\right] = \underbrace{\mathbb{E}\left[\sum_{t=1}^{n} \mathbb{1}(I(t) = a), \overline{\phi}_{a,t} \geq \mu_* - \epsilon_1, D_L(\widehat{F}_{I(t)}, F_{I(t)}) \leq \epsilon_2\right]}_{B1}$$

$$+ \underbrace{\mathbb{E}\left[\sum_{t=1}^{n} \mathbb{1}(I(t) = a), \overline{\phi}_{a,t} < \mu_* - \epsilon_1, D_L(\widehat{F}_{I(t)}, F_{I(t)}) > \epsilon_2\right]}_{B2}$$

We can use the same ways of derivations as in Lemma 5, equation (4) to have the same bound

$$B1 \leq m_0 + \sum_{t=1}^{n} \sum_{m=m_0}^{n} \mathbf{Pr}\left(\overline{\theta}_k(t) \geq \mu_\star - \epsilon_1 \middle| D_L\left(\widehat{F}_a(t), F_a(t)\right) \leq \epsilon_2, T_k(t) = m\right)$$

$$\times \mathbf{Pr}\left(D_L\left(\widehat{F}_a(t), F_a(t)\right) \leq \epsilon_2, T_k(t) = m\right) \tag{7}$$

According to Lemma 15 in Appendix G.1 (29) on conditional probabilities, for any $\nu \in (0, 1)$ we have

$$\mathbf{Pr}\left(\overline{\theta}_k(t) \geq \mu_\star - \epsilon_1 \middle| D_L\left(\widehat{F}_a(t), F_a(t)\right) \leq \epsilon_2, T_k(t) = m\right)$$

$$\leq \frac{1}{\nu} \exp\left\{-n\left(\mathcal{K}_{\text{inf}}(\widehat{F}_a(t), \mu_\star - \epsilon_1) - \nu\frac{\mu_\star - \epsilon_1}{1 - (\mu_\star - \epsilon_1)}\right)\right\}$$

Because $\mathcal{K}_{\text{inf}}(F, \mu)$ is continuous in $F$ with respect to the Levy distance from (19), Theorem 7, for any $\epsilon_3 > 0$ there exists $\epsilon_2 > 0$ such that

$$D_L(\widehat{F}_a(t), F_a) \leq \epsilon_2 \Rightarrow \left|\mathcal{K}_{\text{inf}}(\widehat{F}_a(t), \mu_\star - \epsilon_1) - \mathcal{K}_{\text{inf}}(F_a, \mu_\star - \epsilon_1)\right| \leq \epsilon_3$$

Therefore, $\forall \nu \in (0, 1)$ and for any $\epsilon_5 > 0$, there exists $\epsilon_1, \epsilon_2 > 0$ such that

$$\mathbf{Pr}\left(\overline{\theta}_k(t) \geq \mu_\star - \epsilon_1 \middle| D_L\left(\widehat{F}_a(t), F_a(t)\right) \leq \epsilon_2, T_k(t) = m\right)$$

$$\leq \frac{1}{\nu}\left(-m\left(\mathcal{K}_{\text{inf}}(F_a, \mu_\star - \epsilon_1) - \epsilon_3 - \nu\frac{\mu_\star - \epsilon_1}{1 - (\mu_\star - \epsilon_1)}\right)\right)$$

$$\overset{\text{(Theorem 6 (19))}}{\leq} \frac{1}{\nu}\left(-m\left(\mathcal{K}_{\text{inf}}(F_a, \mu_\star)\frac{\epsilon_1}{1 - \mu_\star} - \epsilon_3 - \nu\frac{\mu_\star - \epsilon_1}{1 - (\mu_\star - \epsilon_1)}\right)\right)$$

This implies that $\forall \epsilon_0 > 0$, there exists $\nu \in (0, 1), \epsilon_1 > 0$ and $\epsilon_2 > 0$ such that

$$\mathbf{Pr}\left(\overline{\theta}_k(t) \geq \mu_\star - \epsilon_1 \middle| D_L\left(\widehat{F}_a(t), F_a(t)\right) \leq \epsilon_2, T_k(t) = m\right) \leq \frac{1}{\nu} \exp\left\{-m(\mathcal{K}_{\text{inf}}(F_a, \mu_\star) - \epsilon_0)\right\}$$

Therefore, according to inequality (7) and the fact that

$$\mathbf{Pr}\left(D_L\left(\widehat{F}_a(t), F_a(t)\right) \leq \epsilon_2, T_k(t) = m\right) \leq 1$$

we have

$$B1 \leq m_0 + \sum_{t=1}^{n} \frac{1}{\nu} \exp\left\{-m(\mathcal{K}_{\text{inf}}(F_a, \mu_\star) - \epsilon_0)\right\}$$

$$\leq m_0 + \frac{1}{\nu}T\exp\left\{-m_0(\mathcal{K}_{\text{inf}}(F_a, \mu_\star) - \epsilon_0)\right\}$$

596 Choose $m_0 = \frac{\log n}{\mathcal{K}_{\inf}(F_a, \mu_\star) - \epsilon_0}$ we have

$$B1 \leq \frac{\log n}{\mathcal{K}_{\inf}(F_a, \mu_\star) - \epsilon_0} + \frac{1}{\nu}$$

597 Also According to Proposition 10 (29), for any $\epsilon_0 > 0$ we have

$$B2 \leq O(1)$$

598 That leads us to

$$\mathbb{E}[T_a(n)] \leq \frac{\log n}{\mathcal{K}_{\inf}(F_a, \mu_\star) - \epsilon_0} + o(\log n) + O(1),$$

599 which concludes the proof. $\qquad\square$

600 **Lemma 7.** *Consider Categorical Thompson Sampling(CATS) strategy applied to a non-stationary*
601 *problem where the pay-off sequence satisfies Assumption 1. Let us define the power mean estimator*
602 $\widehat{\mu}_n(p)$ *as* $\widehat{\mu}_n(p) = \left( \sum_{a=1}^{K} \frac{T_a(n)}{n} \widehat{\mu}_{a,T_a(n)}^p \right)^{\frac{1}{p}}$, *and* $\delta_{\star,n} = \mu_\star - \mu_{\star,n}$ *For any* $p \geq 1, \epsilon_0 > 0$, *we have*

$$|\mathbb{E}[\widehat{\mu}_n(p)] - \mu_\star| \leq |\delta_{\star,n}| + \frac{R}{n} \sum_{a=1, a \neq a_*}^{K} \left\{ \frac{(1 + \epsilon_0) \log n}{\mathcal{K}^{(N)}(F_a, \mu^\star)} + o(\log n) + O(1) \right\}$$

603 *Proof.* We observe that

$$|\widehat{\mu}_n(p) - \mu_\star| \leq |\widehat{\mu}_n(p) - \mu_{\star,n}| + |\mu_\star - \mu_{\star,n}| = |\widehat{\mu}_n(p) - \mu_{\star,n}| + |\delta_{\star,n}|$$

604 Furthermore,

$$\widehat{\mu}_{a,T_a(n)} \leq \mu_{a,n} + |\widehat{\mu}_{a,T_a(n)} - \mu_{a,n}|. \tag{8}$$

605 Since $\mu_{\star,n} = \max_{a \in [K]} \{\mu_{a,n}\}$, we have

$$\widehat{\mu}_n(p) - \mu_{\star,n} = \widehat{\mu}_n(p) - \sum_{a=1}^{K} T_a(n) \mu_{\star,n} \leq \left( \sum_{a=1}^{K} \frac{T_a(n)}{n} \left( \widehat{\mu}_{a,T_a(n)} \right)^p \right)^{\frac{1}{p}} - \left( \sum_{a=1}^{K} \frac{T_a(n)}{n} \left( \mu_{a,n} \right)^p \right)^{\frac{1}{p}}$$

$$= \frac{\left( \sum_{a=1}^{K} T_a(n) \left( \widehat{\mu}_{a,T_a(n)} \right)^p \right)^{\frac{1}{p}} - \left( \sum_{a=1}^{K} T_a(n) \left( \mu_{a,n} \right)^p \right)^{\frac{1}{p}}}{n^{\frac{1}{p}}}$$

606 Applying Minkowski's inequality from Lemma 3, and the result of (8), we have

$$\widehat{\mu}_n(p) - \mu_{\star,n} \leq \frac{\left( \sum_{a=1}^{K} T_a(n) \left( \mu_a + |\widehat{\mu}_{a,T_a(n)} - \mu_{a,n}| \right)^p \right)^{\frac{1}{p}} - \left( \sum_{a=1}^{K} T_a(n) \left( \mu_{a,n} \right)^p \right)^{\frac{1}{p}}}{n^{\frac{1}{p}}}$$

$$\leq \frac{\left( \sum_{a=1}^{K} T_a(n) \left( |\widehat{\mu}_{a,T_a(n)} - \mu_{a,n}| \right)^p \right)^{\frac{1}{p}}}{n^{\frac{1}{p}}}$$

607 On the other hand,

$$\mu_{\star,n} - \widehat{\mu}_n(p) = \frac{n\mu_{\star,n} - n\widehat{\mu}_n(p)}{n} = \frac{n\mu_{\star,n} - \left( \sum_{a=1}^{K} T_a(n) \mu_{a,n} \right) + \sum_{a=1}^{K} T_a(n) \mu_{a,n} - n\widehat{\mu}_n(p)}{n}$$

$$= \frac{\sum_{a=1, a \neq a_*}^{K} T_a(n) |\mu_{\star,n} - \mu_{a,n}| + \sum_{a=1}^{K} T_a(n) \mu_{a,n} - n\widehat{\mu}_n(p)}{n}$$

$$\leq R \sum_{a=1, a \neq a_*}^{K} \frac{T_a(n)}{n} + \sum_{a=1}^{K} \frac{T_a(n)}{n} \mu_{a,n} - \widehat{\mu}_n(p) \tag{9}$$

608 Because power mean is an increasing function of $p$, so that

$$\sum_{a=1}^{K} \frac{T_a(n)}{n} \mu_{a,n} \leq \left( \sum_{a=1}^{K} \frac{T_a(n)}{n} \left( \mu_{a,n} \right)^p \right)^{1/p}.$$

609 Furthermore, we observe that

$$\mu_{a,n} \leq \widehat{\mu}_{a,T_a(n)} + \left| \widehat{\mu}_{a,T_a(n)} - \mu_{a,n} \right|.$$

610 So that, from equation (9) we have

$$\mu_{\star,n} - \widehat{\mu}_n(p) \leq R \sum_{a=1,a\neq a_*}^{K} \frac{T_a(n)}{n} + \left( \sum_{a=1}^{K} \frac{T_a(n)}{n} \left( \mu_{a,n} \right)^p \right)^{1/p} - \widehat{\mu}_n(p)$$

$$\leq R \sum_{a=1,a\neq a_*}^{K} \frac{T_a(n)}{n}$$

$$+ \frac{\left( \sum_{a=1}^{K} T_a(n) \left( \widehat{\mu}_{a,T_a(n)} + \left| \widehat{\mu}_{a,T_a(n)} - \mu_{a,n} \right| \right)^p \right)^{\frac{1}{p}} - \left( \sum_{a=1}^{K} T_a(n) \left( \widehat{\mu}_{a,T_a(n)} \right)^p \right)^{\frac{1}{p}}}{n^{\frac{1}{p}}}$$

$$\overset{\text{(Minkovski's inequality)}}{\leq} R \sum_{a=1,a\neq a_*}^{K} \frac{T_a(n)}{n} + \frac{\left( \sum_{a=1}^{K} T_a(n) \left( \left| \widehat{\mu}_{a,T_a(n)} - \mu_{a,n} \right| \right)^p \right)^{\frac{1}{p}}}{n^{\frac{1}{p}}}$$

$$\overset{\text{(Properties of } L^p \text{ norm)}}{\leq} R \sum_{a=1,a\neq a_*}^{K} \frac{T_a(n)}{n} + \frac{\left( \sum_{a=1}^{K} T_a(n) \left( \left| \widehat{\mu}_{a,T_a(n)} - \mu_{a,n} \right| \right) \right)}{n^{\frac{1}{p}}}$$

$$= R \sum_{a=1,a\neq a_*}^{K} \frac{T_a(n)}{n} + \frac{\sum_{a=1}^{K} \left( \left| \sum_{t}^{T_a(n)} R_{a,t} - T_a(n)\mu_{a,n} \right| \right)}{n^{\frac{1}{p}}}$$

611 Therefore

$$\left| \mathbb{E}[\widehat{\mu}_n(p) - \mu_{\star,n}] \right| \leq R \sum_{a=1,a\neq a_*}^{K} \frac{\mathbb{E}[T_a(n)]}{n} + \frac{\mathbb{E}\left[ \left( \left| \sum_{a=1}^{K} \sum_{t}^{T_a(n)} R_{a,t} - T_a(n)\mu_{a,n} \right| \right) \right]}{n^{\frac{1}{p}}}$$

$$= R \sum_{a=1,a\neq a_*}^{K} \frac{\mathbb{E}[T_a(n)]}{n}$$

612 Please note that because we study non-stationary bandits, $\mathbb{E}[\sum_{t}^{n} R_{a,t}] = n\mu_{a,n}$, therefore,

$$\frac{\mathbb{E}\left[ \left( \left| \sum_{a=1}^{K} \sum_{t}^{T_a(n)} R_{a,t} - T_a(n)\mu_{a,n} \right| \right) \right]}{n^{\frac{1}{p}}} = 0$$

613 According to Lemma 5, we have

$$\left| \mathbb{E}[\widehat{\mu}_n(p) - \mu_{\star,n}] \right| \leq R \sum_{a=1,a\neq a_*}^{K} \frac{\mathbb{E}[T_a(n)]}{n} \leq \frac{R}{n} \sum_{a=1,a\neq a_*}^{K} \left\{ \frac{(1+\epsilon_0)\log n}{\mathcal{K}^{(N)}(F_a, \mu^\star)} + o(\log n) + O(1) \right\},$$

614 which concludes the proof. □

615 **Lemma 8.** *Consider Particle Thompson Sampling(PATS) strategy applied to a non-stationary*
616 *problem where the pay-off sequence satisfies Assumption 1. Let us define the power mean estimator*
617 $\widehat{\mu}_n(p)$ *as* $\widehat{\mu}_n(p) = \left( \sum_{a=1}^{K} \frac{T_a(n)}{n} \widehat{\mu}_{a,T_a(n)}^p \right)^{\frac{1}{p}}$, *and* $\delta_{\star,n} = \mu_\star - \mu_{\star,n}$ *For any* $p \geq 1, \epsilon_0 > 0$, *we have*

$$\left| \mathbb{E}[\widehat{\mu}_n(p)] - \mu_\star \right| \leq |\delta_{\star,n}| + \frac{R}{n} \sum_{a=1,a\neq a_*}^{K} \left\{ \frac{\log n}{\mathcal{K}_{inf}(F_a, \mu^\star) - \epsilon_0} + o(\log n) + O(1) \right\}$$

618 *Proof.* Similar to Lemma 7, we can derive

$$|\mathbb{E}[\widehat{\mu}_n(p) - \mu_{\star,n}]| \leq |\delta_{\star,n}| + R \sum_{a=1,a\neq a_*}^{K} \frac{\mathbb{E}[T_a(n)]}{n}.$$

619 And according to Lemma 6, we have

$$|\mathbb{E}[\widehat{\mu}_n(p) - \mu_{\star,n}]| \leq R \sum_{a=1,a\neq a_*}^{K} \frac{\mathbb{E}[T_a(n)]}{n} \leq \frac{R}{n} \sum_{a=1,a\neq a_*}^{K} \left\{ \frac{\log n}{\mathcal{K}_{\inf}(F_a,\mu^\star) - \epsilon_0} + o(\log n) + O(1) \right\},$$

620 which concludes the proof. $\qquad\square$

621 **Theorem 1.** *For $a \in [K]$, let $(\widehat{\mu}_{a,n})_{n\geq 1}$ be a sequence of estimator satisfying $\underset{n\to\infty}{plim}\,\widehat{\mu}_{a,n} = \mu_a$ and*

622 *let $\mu_\star = \underset{a}{\max}\{\mu_a\}$. Assume that all the estimators are bounded in $[0, R]$. We consider a bandit*

623 *algorithm that selects each arm according to CATS once in each round $n \geq K$.*

624 *Then, for all $p \in [1, \infty)$, the sequence of estimators*

$$\widehat{\mu}_n(p) = \left( \sum_{a=1}^{K} \frac{T_a(n)}{n} \widehat{\mu}_{a,T_a(n)}^{p} \right)^{\frac{1}{p}},$$

625 *where $T_a(n) = \sum_{t=1}^{n-1} \mathbb{1}(a_t = a)$ is the number of selections of $a$ prior to round $n$ satisfies*

$$\underset{n\to\infty}{plim}\,\widehat{\mu}_n(p) = \mu_\star.$$

626 *Proof.* We first prove that $\lim_{n\to\infty} \mathbb{E}[\widehat{\mu}_n(p)] = \mu_*$. According to the result of Lemma 7, we have

$$|\mathbb{E}[\widehat{\mu}_n(p)] - \mu_\star| \leq |\delta_{\star,n}| + R \sum_{a=1,a\neq a_*}^{K} \frac{\mathbb{E}[T_a(n)]}{n}$$

$$\leq |\delta_{\star,n}| + \frac{R}{n} \sum_{a=1,a\neq a_*}^{K} \left\{ \frac{(1+\epsilon_0)\log n}{\mathcal{K}^{(N)}(F_a,\mu^\star)} + o(\log n) + O(1) \right\}$$

627 with $\delta_{\star,n} = \mu_\star - \mu_{\star,n}$, and because $\lim_{n\to\infty} \mu_{*,n} = \mu_\star$, we can concludes that

$$\lim_{n\to\infty} \mathbb{E}[\widehat{\mu}_n(p)] = \mu_*.$$

628 Second, we prove that

$$\forall n \geq 1, \forall \varepsilon > 0, \exists c > 0 \text{ that } \mathbb{P}\left(|\widehat{\mu}_n(p) - \mu_\star| > \varepsilon\right) \leq cn^{-1}\varepsilon^{-1}.$$

629 We observe that

$$|\widehat{\mu}_n(p) - \mu_\star| \leq |\widehat{\mu}_n(p) - \mu_{\star,n}| + |\mu_\star - \mu_{\star,n}| = |\widehat{\mu}_n(p) - \mu_{\star,n}| + |\delta_{\star,n}|$$
$$\Longrightarrow \mathbb{P}(|\widehat{\mu}_n(p) - \mu_\star| \geq \epsilon) \leq \mathbb{P}(|\widehat{\mu}_n(p) - \mu_{\star,n}| \geq \epsilon/2) + \mathbb{P}(|\delta_{\star,n}| \geq \epsilon/2).$$

630 Because $\lim_{n\to n} |\delta_{\star,n}| = 0$, therefore, $\exists N_0 > 0$ such that $\forall n \geq N_0$, we have $|\delta_{\star,n}| < \epsilon/2$ that means

$$\forall n > N_0, \mathbb{P}(|\delta_{\star,n}| \geq \epsilon/2) = 0.$$

631 Next, according to Lemma 7,

$$|\mathbb{E}[\widehat{\mu}_n(p)] - \mu_{\star,n}| \leq \frac{R}{n} \sum_{a=1,a\neq a_*}^{K} \left\{ \frac{(1+\epsilon_0)\log n}{\mathcal{K}^{(N)}(F_a,\mu^\star)} + o(\log n) + O(1) \right\} = O(n^{-1}),$$

632 that leads to

$$\mathbb{P}(|\widehat{\mu}_n(p) - \mu_{\star,n}| \geq \epsilon/2) \leq \frac{|\mathbb{E}[\widehat{\mu}_n(p)] - \mu_{\star,n}|}{\epsilon/2} = \frac{O(n^{-1})}{\epsilon/2}.$$

633 Therefore, $\exists c > 0$ such that

$$\mathbb{P}(|\widehat{\mu}_n(p) - \mu_{\star,n}| \geq \epsilon/2) \leq cn^{-1}\epsilon^{-1},$$

634 which means

$$\forall n \geq N_0, \forall \varepsilon > 0, \exists c > 0 \text{ that } \mathbb{P}\left(|\widehat{\mu}_n(p) - \mu_\star| > \varepsilon\right) \leq cn^{-1}\varepsilon^{-1}.$$

635 Now we see that $|\widehat{\mu}_n(p) - \mu_\star| \leq R$. With $\epsilon \geq R$, we have $|\widehat{\mu}_n(p) - \mu_\star| > \epsilon \Leftrightarrow |\widehat{\mu}_n(p) - \mu_\star| > R$,
636 therefore the inequality holds as

$$\mathbb{P}\left(|\widehat{\mu}_n(p) - \mu_\star| > \varepsilon\right) = 0 \leq cn^{-1}\varepsilon^{-1}.$$

637 with $0 < \epsilon < R, 1 \leq n < N_0 \Rightarrow n\epsilon < RN_0 \Rightarrow n^{-1}\varepsilon^{-1} > 1/RN_0$. Therefore

$$\forall C > 1/RN_0 \Rightarrow \mathbb{P}\left(|\widehat{\mu}_n(p) - \mu_\star| > \varepsilon\right) \leq 1 < Cn^{-1}\varepsilon^{-1},$$

638 which means

$$\forall n \geq 1, \forall \varepsilon > 0, \exists C > 0 \text{ that } \mathbb{P}\left(|\widehat{\mu}_n(p) - \mu_\star| > \varepsilon\right) \leq Cn^{-1}\varepsilon^{-1}.$$

639 That concludes the proof. $\qquad\qquad\square$

640 **Theorem 2.** *For $a \in [K]$, let $(\widehat{\mu}_{a,n})_{n \geq 1}$ be a sequence of estimator satisfying $\underset{n \to \infty}{plim}\, \widehat{\mu}_{a,n} = \mu_a$ and*
641 *let $\mu_\star = \underset{a}{\max}\{\mu_a\}$. Assume that all the estimators are bounded in $[0, R]$. We consider a bandit*
642 *algorithm that selects each arm according to PATS once in each round $n \geq K$.*

643 *Then, for all $p \in [1, \infty)$, the sequence of estimators*

$$\widehat{\mu}_n(p) = \left( \sum_{a=1}^{K} \frac{T_a(n)}{n} \widehat{\mu}_{a,T_a(n)}^p \right)^{\frac{1}{p}},$$

644 *where $T_a(n) = \sum_{t=1}^{n-1} \mathbb{1}(a_t = a)$ is the number of selections of $a$ prior to round $n$ satisfies*

$$\underset{n \to \infty}{plim}\, \widehat{\mu}_n(p) = \mu_\star.$$

645 *Proof.* The proof follows the same steps as Theorem 1. We first prove that $\lim_{n \to \infty} \mathbb{E}[\widehat{\mu}_n(p)] = \mu_*$.
646 According to the result of Lemma 8, we have

$$|\mathbb{E}[\widehat{\mu}_n(p)] - \mu_\star| \leq |\delta_{\star,n}| + R \sum_{a=1,a\neq a_*}^{K} \frac{\mathbb{E}[T_a(n)]}{n}$$

$$\leq |\delta_{\star,n}| + \frac{R}{n} \sum_{a=1,a\neq a_*}^{K} \left\{ \frac{\log n}{\mathcal{K}_{\inf}(F_a, \mu^\star) - \epsilon_0} + o(\log n) + O(1) \right\}$$

647 with $\delta_{\star,n} = \mu_\star - \mu_{\star,n}$, and because $\lim_{n \to \infty} \mu_{*,n} = \mu_\star$, we can concludes that

$$\lim_{n \to \infty} \mathbb{E}[\widehat{\mu}_n(p)] = \mu_*.$$

648 Second, we prove that

$$\forall n \geq 1, \forall \varepsilon > 0, \exists c > 0 \text{ that } \mathbb{P}\left(|\widehat{\mu}_n(p) - \mu_\star| > \varepsilon\right) \leq cn^{-1}\varepsilon^{-1}.$$

649 We observe that

$$|\widehat{\mu}_n(p) - \mu_\star| \leq |\widehat{\mu}_n(p) - \mu_{\star,n}| + |\mu_\star - \mu_{\star,n}| = |\widehat{\mu}_n(p) - \mu_{\star,n}| + |\delta_{\star,n}|$$

$$\Longrightarrow \mathbb{P}(|\widehat{\mu}_n(p) - \mu_\star| \geq \epsilon) \leq \mathbb{P}(|\widehat{\mu}_n(p) - \mu_{\star,n}| \geq \epsilon/2) + \mathbb{P}(|\delta_{\star,n}| \geq \epsilon/2).$$

650 Because $\lim_{n \to n} |\delta_{\star,n}| = 0$, therefore, $\exists N_0 > 0$ such that $\forall n \geq N_0$, we have $|\delta_{\star,n}| < \epsilon/2$ that means

$$\forall n > N_0, \mathbb{P}(|\delta_{\star,n}| \geq \epsilon/2) = 0.$$

Next, according to Lemma 8,

$$|\mathbb{E}[\widehat{\mu}_n(p)] - \mu_{\star,n}| \leq \frac{R}{n} \sum_{a=1, a \neq a_*}^{K} \left\{ \frac{\log n}{\mathcal{K}_{\inf}(F_a, \mu^\star) - \epsilon_0} + o(\log n) + O(1) \right\} = O(n^{-1}),$$

that leads to

$$\mathbb{P}(|\widehat{\mu}_n(p) - \mu_{\star,n}| \geq \epsilon/2) \leq \frac{|\mathbb{E}[\widehat{\mu}_n(p)] - \mu_{\star,n}|}{\epsilon/2} = \frac{O(n^{-1})}{\epsilon/2}.$$

Therefore, $\exists c > 0$ such that

$$\mathbb{P}(|\widehat{\mu}_n(p) - \mu_{\star,n}| \geq \epsilon/2) \leq cn^{-1}\epsilon^{-1},$$

which means

$$\forall n \geq N_0, \forall \varepsilon > 0, \exists c > 0 \text{ that } \mathbb{P}\left(|\widehat{\mu}_n(p) - \mu_\star| > \varepsilon\right) \leq cn^{-1}\varepsilon^{-1}.$$

Now we see that $|\widehat{\mu}_n(p) - \mu_\star| \leq R$. With $\epsilon \geq R$, we have $|\widehat{\mu}_n(p) - \mu_\star| > \epsilon \Leftrightarrow |\widehat{\mu}_n(p) - \mu_\star| > R$, therefore the inequality holds as

$$\mathbb{P}\left(|\widehat{\mu}_n(p) - \mu_\star| > \varepsilon\right) = 0 \leq cn^{-1}\varepsilon^{-1}.$$

with $0 < \epsilon < R, 1 \leq n < N_0 \Rightarrow n\epsilon < RN_0 \Rightarrow n^{-1}\varepsilon^{-1} > 1/RN_0$. Therefore

$$\forall C > 1/RN_0 \Rightarrow \mathbb{P}\left(|\widehat{\mu}_n(p) - \mu_\star| > \varepsilon\right) \leq 1 < Cn^{-1}\varepsilon^{-1},$$

which means

$$\forall n \geq 1, \forall \varepsilon > 0, \exists C > 0 \text{ that } \mathbb{P}\left(|\widehat{\mu}_n(p) - \mu_\star| > \varepsilon\right) \leq Cn^{-1}\varepsilon^{-1}.$$

That concludes the proof. $\qquad\square$

# E   Convergence of CATS and PATS in Monte-Carlo Tree Search

Based upon the results of CATS and PATS using power mean as the value backup operator on the described non-stationary multi-armed bandit problem, we derive theoretical results for CATS in an MCTS tree.

We derive Theorem 3 for CATS and Theorem 4 for PATS, which show concentration and convergence for any internal node in the tree. These proofs utilize induction, leveraging the results of Lemma 7 for CATS and Lemma 8 for PATS, and Lemma 5 for CATS and Lemma 6 for PATS. Additionally, we use Lemma 1, which demonstrates the concentration and convergence of an estimated Q-value based on the child V-value node, applying it recursively throughout the tree.

Our main results, Theorem 5 for CATS and Theorem 5 for PATS, show that the simple regret converges non-asymptotically at a rate of $O(n^{-1})$.

**Theorem 3.** *When we apply the CATS algorithm, we have*
*(i) For any node $s_h$ at the depth $h^{th}$ in the tree,*

$$\plim_{n \to \infty} \widehat{Q}_n(s_h, a_k) = \widetilde{Q}(s_h, a_k).$$

*(ii) For any node $s_h$ at the depth $h^{th}$ in the tree,*

$$\plim_{n \to \infty} \widehat{V}_n(s_h) = \widetilde{V}(s_h).$$

*Proof.* We will prove this by induction on the depth $D$ of the tree. If the tree only has depth $(1)$. The state at the root node is $s_0$, let us assume that at time step $t$, after taking action $a_k$, the MCTS tree gets an intermediate reward $r_t(s_0, a_k)$ and traverses to the next state $s_1$. Let us assume that $R(s_0, a_k)$ is the mean of the intermediate reward at state $s_0$, after taking action $a_k$. We recall the definition of $\widetilde{Q}(s_0, a_k)$, with $\pi_0$ is the rollout policy to estimate the newly added node at the leaf,

$$\widetilde{Q}(s_0, a_k) = R(s_0, a_k) + \gamma \sum_{s_1 \in \mathcal{A}_{s_0}} \mathbb{P}(s_1|s_0, a_k)\widetilde{V}(s_1)$$

where $\widetilde{V}(s_1)$ is the value of the policy $\pi_0$ at state $s_1$, $\mathcal{A}_{s_0}$ is the set of feasible actions at state $s_0$, $|\mathcal{A}_{s_0}| = M$, $\mathbb{P}(s_1|s_0, a_k)$ is the probability transition of taking action $a_k$ at state $s_0$ to state $s_1$. From ((1)), we have

$$\widehat{Q}_n(s_0, a_k) = \frac{1}{n} \sum_{t=1}^{n} r_t(s_0, a_k) + \gamma \sum_{s_1 \sim \tau(s_0, a_k)} \frac{T_{s_0, a_k}^{s_1}(n)}{n} \widehat{V}_{T_{s_0, a_k}^{s_1}(n)}(s_1)$$

$(i)$ is a direct result of Lemma 1 with $X_t$ is the intermediate reward $r_t(s_0, a_k)$ at time $t$, $p = (p_1, p_2, ...p_M) \sim \mathbb{P}(\cdot|s_0, a_k)$, where $\mathbb{P}(\cdot|s_0, a_k)$ is the probability transition dynamic of taking action $a_k$ at state $s_0$. For $m \in [M]$, each $(\widehat{V}_{m,t})_{t\geq 1}$ at time step t is the deterministic initial Value function $\widetilde{V}(s_1)$. We have

$$\plim_{n \to \infty} \widehat{V}_{m,n}(s_1) = \widetilde{V}(s_1), \text{ with } s_1 \in \{s_m\}, m = 1, 2, 3...M, \text{ where } s_m \sim \tau(\cdot|s_0, a_k)$$

$(ii)$ Direct results from Theorem 1. In detail, we have from $(i)$,

$$\plim_{n \to \infty} \widehat{Q}_n(s_0, a_k) = \widetilde{Q}(s_0, a_k), \text{ with } a_k \in \mathcal{A}_{s_0}$$

Because by definition:

$$\widetilde{V}(s_0) = \max_{a_k \in \mathcal{A}_{s_0}} \widetilde{Q}(s_0, a_k)$$

$$\widehat{V}_n(s_0) = \left( \sum_{a \in \mathcal{A}_{s_0}} \frac{T_{s_0, a}(n)}{n} \left( \widehat{Q}_{T_{s_0, a}(n)}(s_0, a) \right)^p \right)^{\frac{1}{p}} \quad \text{for some } p \in [1, +\infty)$$

Then we have

$$\plim_{n \to \infty} \widehat{V}_n(s_0) = \widetilde{V}(s_0)$$

that concludes for $(ii)$

Let us assume that with the tree of depth $D$, the theorem holds for all its children.

Now let's consider the tree with depth $(D + 1)$. When we take one action at the root node at the state $s_0$, it comes to a subtree with depth $(D)$. According to the induction assumption, the results hold for any internal node in the tree after we take the first action. We have $s_1 \sim \tau(s_0, a_k)$. By the definition, $\widetilde{V}(s_H) = V_0(s_H)$ and, for all $h \leq H - 1$,

$$\widetilde{Q}(s_h, a) = R(s_h, a) + \gamma \sum_{s_{h+1} \in \mathcal{A}_s} \mathbb{P}(s_{h+1}|s_h, a)\widetilde{V}(s_{h+1})$$

$$\widetilde{V}(s_h) = \max_a \widetilde{Q}(s_h, a)$$

By the assumption of the induction the root node of a subtree with depth $(D)$ at state $s_1$ we have

$$\plim_{n \to \infty} \widehat{V}_n(s_1) = \widetilde{V}(s_1)$$

$(i)$ Let's apply Lemma 1 with $\{X_t\}$ is the intermediate reward $\{r_t(s_0, a_k)\}$, $p = (p_1, p_2, ...p_M) \sim \mathbb{P}(\cdot|s_0, a_k)$. For $m \in [M]$, each $(\widehat{V}_{m,t})_{t\geq 1}$ at time step t is the empirical Value function $\widehat{V}_t(s_1)$. We will have

$$\plim_{n \to \infty} \widehat{Q}_n(s_0, a_k) = \widetilde{Q}(s_0, a_k), \text{ with } a_k \in \mathcal{A}_{s_0}$$

$(ii)$ follows the results of Theorem 1 as at the root node $s_0$ of depth $D + 1$, with

$$\widetilde{V}(s_0) = \max_{a_k \in \mathcal{A}_{s_0}} \widetilde{Q}(s_0, a_k)$$

$$\widehat{V}_n(s_0) = \left( \sum_{a \in \mathcal{A}_s} \frac{T_{s_0, a}(n)}{n} \left( \widehat{Q}_{T_{s_0, a}(n)}(s_0, a) \right)^p \right)^{\frac{1}{p}} \quad \text{for some } p \in [1, +\infty)$$

701  And because

$$\operatorname*{plim}_{n\to\infty}\widehat{Q}_n(s_0, a_k) = \widetilde{Q}(s_0, a_k), \text{ with } a_k \in \mathcal{A}_{s_0}$$

702  Then, we have

$$\operatorname*{plim}_{n\to\infty}\widehat{V}_n(s_0) = \widetilde{V}(s_0).$$

703  that concludes for $(ii)$

704  The results of Theorem 3 hold for any node in the tree with the tree of depth $(D+1)$. By induction,
705  we can conclude the proof. $\qquad\square$

706  Similarly we can derive the following Theorem

707  **Theorem 4.** *When we apply the PATS algorithm, we have*
708      *(i) For any node $s_h$ at the depth $h^{th}$ in the tree,*

$$\operatorname*{plim}_{n\to\infty}\widehat{Q}_n(s_h, a_k) = \widetilde{Q}(s_h, a_k).$$

709      *(ii) For any node $s_h$ at the depth $h^{th}$ in the tree,*

$$\operatorname*{plim}_{n\to\infty}\widehat{V}_n(s_h) = \widetilde{V}(s_h).$$

711  *Proof.* The proof follows the same steps as Theorem 3 by applying the results of Lemma 1 and
712  Theorem 2. $\qquad\square$

713  **Theorem 5.** *(Convergence of Expected Payoff of CATS) We have at the root node $s_0$,*

$$\mathbb{E}\left[\left|\widehat{V}_n(s_0) - V^\star(s_0)\right|\right] \le O(n^{-1}).$$

714  *Proof.* We prove the result by induction and use the results of Theorem 3 to prove this Theorem. Let
715  us assume that the depth of the tree is $D = 1$, as the results of Lemma 7, we have

$$\left|\mathbb{E}[\widehat{V}_n(s_0)] - V^\star(s_0)\right| \le |\delta_{\star,n}| + O(\frac{\log n}{n}) = |\delta_{\star,n}| + O(n^{-1}).$$

716  And because the tree only have the depth $D = 1$, we have $|\delta_{\star,n}| = 0$, so that the result holds at
717  the depth $D = 1$. Let us assume that we have the result of the tree at the depth $D$. Now when the
718  depth of the tree is $D + 1$, at the root node $s_0$, the conditions of Assumption 1 hold as the results of
719  Theorem 3 then we have

$$\left|\mathbb{E}[\widehat{V}_n(s_0)] - V^\star(s_0)\right| \overset{\text{(Lemma 7)}}{\le} |\delta_{\star,n}| + O(\frac{\log n}{n}) = |\delta_{\star,n}| + O(n^{-1}),$$

720  where the bias

$$|\delta_{\star,n}| = \left|\mathbb{E}[\widehat{Q}_n(s_0, a_\star)] - Q^\star(s_0, a_\star)\right| \overset{\text{(contraction)}}{\le} \gamma \parallel \mathbb{E}[\widehat{V}_n^{(1)}] - V^\star \parallel_\infty \overset{\text{(by induction)}}{\le} \gamma O(n^{-1}).$$

721  Therefore,

$$\left|\mathbb{E}[\widehat{V}_n(s_0)] - V^\star(s_0)\right| \le O(n^{-1}),$$

722  that concludes the proof. $\qquad\square$

723  Next, we present the results of Theorem 6. The proof follows the same steps as Theorem 5.

724  **Theorem 6.** *(Convergence of Expected Payoff of PATS) We have at the root node $s_0$,*

$$\mathbb{E}\left[\left|\widehat{V}_n(s_0) - V^\star(s_0)\right|\right] \le O(n^{-1}).$$

# F  Limitations

**Computational Demands**: The CATS distributional Monte Carlo Tree Search (MCTS) faces challenges in managing computational demands while maintaining and updating probability distributions, leading to a slightly increased complexity.

**Fixed precision**: The PATS set of particles can increase in size if the observed value are different. We prevent this in the implementation by fixing the float precision.

**Number of atoms**: Our approach's performance is slightly influenced by hyperparameters, with the number of atoms being a critical factor. Suboptimal choices may affect performance.

# G  Experimental setup

All the experiments were done on 8 Intel Xeon Gold 6130 (Skylake), x86_64, 2.10GHz, 2 CPUs/node, 16 cores/CPU. Whenever feasible, we opted for open-source implementations of algorithms and environments.

**Parameters selection** We search the number of atoms from $\{10,20,...,100\}$ and choose the results with best performances. We set the discount factor $\gamma = .99$ for MDPs, and $\gamma = .95$ for POMDPs. For UCT, we use the exploration constant $C = \sqrt{2} \times (R_{\max} - R_{\min})$.

**Atari hyperparameters** We run CATS in Atari with 10 random seeds, where each seed with 512 samples and collect the average score. We found that only 512 simulations were necessary due to the utilization of a pretrained neural network. We run CATS with 100 atoms. The temperature parameter $\tau$ of MENTS and TENTS is tuned from $\{0.01, 0.02, 0.03, 0.04, 0.05, 0.06, 0.07, 0.08, 0.09, 0.1, 0.2, 0.3, 0.4, 0.5, 0.6, 0.7, 0.8, 0.9, 1.0\}$. The selected parameter $\tau$ are shown in Table 4. The exploration constant $\epsilon$ for MENTS and TENTS are set to $0.01$. For Power-UCT, we select the power mean $p = 2$.

**Atari**

Table 3: Average scores in Atari with 512 samples (10 seeds) $\pm$ 2 times std.

| | CATS | PATS | UCT | DQN | Power-UCT | TENTS | MENTS |
|---|---|---|---|---|---|---|---|
| Phoenix | $3290.00 \pm 1599.52$ | $3619.00 \pm 891.72$ | $2450.00 \pm 786.22$ | $340.0 \pm 0.00$ | $560.00 \pm 0.00$ | $4423.00 \pm 642.38$ | $3098.30 \pm 919.65$ |
| MsPacman | $2058.00 \pm 243.93$ | $2232.00 \pm 896.29$ | $1792.00 \pm 62.85$ | $1930.00 \pm 224.83$ | $1982.00 \pm 473.45$ | $1922.00 \pm 416.91$ | $2018.30 \pm 316.98$ |
| Alien | $1765.0 \pm 801.03$ | $1724.00 \pm 649.63$ | $1900.00 \pm 00.00$ | $1094.00 \pm 122.83$ | $1748.00 \pm 120.21$ | $1613.00 \pm 296.96$ | $1508.60 \pm 322.58$ |
| SpaceInvaders | $826.0 \pm 194.76$ | $791.0 \pm 332.52$ | $525.00 \pm 0.00$ | $525.00 \pm 0.00$ | $672.00 \pm 148.42$ | $742.50 \pm 193.53$ | $832.55 \pm 211.95$ |
| BeamRider | $1952.00 \pm 500.04$ | $1848.0 \pm 320.29$ | $1889.60 \pm 171.09$ | $1952.00 \pm 0.00$ | $1577.60 \pm 112.47$ | $3013.00 \pm 778.89$ | $2822.18 \pm 697.31$ |
| Asterix | $6040.00 \pm 1560.89$ | $5495.00 \pm 3106.64$ | $5380.00 \pm 1464.05$ | $6220.00 \pm 156.80$ | $5540.00 \pm 863.39$ | $5180.00 \pm 528.19$ | $5576.00 \pm 1397.91$ |
| Robotank | $11.50 \pm 2.11$ | $11.9 \pm 1.51$ | $12.2 \pm 1.04$ | $10.20 \pm 0.39$ | $11.00 \pm 1.55$ | $12.10 \pm 1.47$ | $11.59 \pm 1.36$ |
| Seaquest | $3170.00 \pm 787.61$ | $3288.0 \pm 889.41$ | $3564.00 \pm 86.83$ | $2304.00 \pm 531.31$ | $2704.00 \pm 318.93$ | $2928.00 \pm 801.11$ | $3312.40 \pm 390.77$ |
| Solaris | $1062.0 \pm 519.21$ | $1196.00 \pm 524.45$ | $392.00 \pm 198.61$ | $1112.00 \pm 521.53$ | $452.00 \pm 153.19$ | $1168.00 \pm 516.33$ | $1118.20 \pm 513.00$ |
| Asteroids | $930.00 \pm 100.12$ | $953.00 \pm 107.05$ | $5380.00 \pm 1464.05$ | $860.00 \pm 48.89$ | $930.00 \pm 54.66$ | $1518.00 \pm 121.48$ | $1414.70 \pm 261.59$ |
| Enduro | $142.40 \pm 31.21$ | $131.10 \pm 17.16$ | $127.00 \pm 10.07$ | $133.60 \pm 8.73$ | $134.00 \pm 6.69$ | $115.40 \pm 18.82$ | $128.79 \pm 16.26$ |
| Atlantis | $35890.00 \pm 1914.28$ | $36180.0 \pm 2592.70$ | $34300.00 \pm 00.00$ | $34480.00 \pm 119.76$ | $35420.00 \pm 1494.63$ | $36280.00 \pm 1476.24$ | $36277.00 \pm 1811.53$ |
| Hero | $3006.50 \pm 9.16$ | $3020.50 \pm 27.24$ | $3011.50 \pm 17.04$ | $3005.00 \pm 9.53$ | $2998.00 \pm 35.16$ | $3008.00 \pm 0.00$ | $3044.55 \pm 181.04$ |
| Frostbite | $1582.00 \pm 1041.37$ | $1580.00 \pm 1127.23$ | $1900.00 \pm 00.00$ | $2407.00 \pm 116.76$ | $1754.00 \pm 651.38$ | $2357.00 \pm 398.45$ | $2388.20 \pm 320.37$ |
| WizardOfWor | $670.0 \pm 192.09$ | $590.00 \pm 359.02$ | $200.00 \pm 00.00$ | $530.00 \pm 92.63$ | $640.00 \pm 134.53$ | $1210.00 \pm 183.52$ | $1211.00 \pm 314.30$ |
| Breakout | $315.00 \pm 85.80$ | $302.10 \pm 70.47$ | $271.8 \pm 54.63$ | $288.10 \pm 53.01$ | $289.00 \pm 44.46$ | $337.00 \pm 15.91$ | $309.03 \pm 35.13$ |

Atari environments (4) provide diverse video game-inspired scenarios commonly used in reinforcement learning research. These environments offer challenges based on classic Atari 2600 games (23; 38; 6). To explore enhanced exploration in deep reinforcement learning, we employ a Deep Q-Network pre-trained following the experimental setup outlined in (23). This pre-trained network initializes action-values for each node, combined with a Monte-Carlo Tree Search method similar to the AlphaGo one. Here, $P_{prior}$ represents the Boltzmann distribution derived from the action-values $Q(s,.)$ computed by the network. The results in Table 3 show that CATS and PATS outperform UCT, DQN, Power-UCT, TENTS and MENTS in most of the games. For example, CATS is significant better than other methods in *Breakout, Enduro*, while PATS is significant better than other methods in *MsPacman, Solaris*. Our intention in this experiment is not to assert exceptional superiority, but rather to emphasize that CATS and PATS actually work in complicated Atari benchmark.

Table 4: The hyperparameter $\tau$ (temperature) for MENTS and TENTS in Atari.

| | MENTS | TENTS |
|---|---|---|
| Phoenix | 0.07 | 0.6 |
| MsPacman | 0.09 | 0.03 |
| Alien | 0.1 | 0.03 |
| SpaceInvaders | 0.02 | 0.06 |
| BeamRider | 0.02 | 0.03 |
| Asterix | 0.02 | 0.1 |
| Robotank | 0.01 | 0.05 |
| Seaquest | 0.02 | 0.03 |
| Solaris | 0.03 | 0.06 |
| Asteroids | 0.08 | 0.2 |
| Qbert | 0.02 | 0.4 |
| Enduro | 0.02 | 0.1 |
| Atlantis | 0.08 | 0.03 |
| Hero | 0.4 | 0.03 |
| Frostbite | 0.01 | 0.02 |
| WizardOfWor | 0.1 | 0.01 |
| Breakout | 0.02 | 0.04 |

