# OpenReview forum: "Distributional Monte-Carlo Planning with Thompson Sampling in Stochastic Environments"
_NeurIPS.cc/2024/Conference — Submitted to NeurIPS 2024_

### Official Review · Reviewer_Fztp · 2024-07-10

**Soundness:** 2
**Presentation:** 1
**Contribution:** 1
**Rating:** 3
**Confidence:** 4

**Summary:**

The authors explore the use of distributional reinforcement learning within Monte Carlo Tree search. They propose two algorithms CATS and PATS a categorical distribution and particle distribution based approach respectfully. They perform a theoretical analysis of the methods and show analysis of regret. They then evaluate on a synthetic planning tasks and evaluate it in combination with a pre-trained network on the atari benchmark.

**Strengths:**

- Thorough theoretical analysis
- The authors address distributional RL applied to planning which is a clear important direction of research

**Weaknesses:**

- Lack of referencing of existing work and novelty relative to existing work
    - Hayes, C.F., Reymond, M., Roijers, D.M., Howley, E. and Mannion, P., 2023. Monte Carlo tree search algorithms for risk-aware and multi-objective reinforcement learning. *Autonomous Agents and Multi-Agent Systems*, *37*(2), p.26.
- Unjustifiable statement: “For example, CATS is significantly better than other methods in Breakout, Enduro” There is no significance testing performed so this statement cannot be made and in fact the Confidence intervals overlap
- Key results in appendix and lack of empirical results in the main paper
- CATS never outperforms fixed depth MCTS on the synthetic tree task
- Unable to find code despite checklist saying it is provided

- Small issues
    - Figure 2 algorithms alignment off
    - Indication of Atari results in section 5 which are not there
    - Adding bold to best performing method in the Atari table would be useful for readability

**Questions:**

- How does your approach differ to existing methods (Hayes et al)?
- Why do the baselines investigated change between the synthetic experiments and the atari experiments?
- Why do you think CATS is consistently outperformed by other baselines on the synthetic tree task?

**Limitations:**

- Limitations are not included in the main body of the paper which they should be especially considering there is space. The limitations are also not thoroughly discussed for example
    - “faces challenges in managing computational demand” : this does not say anything meaningful
    - “Our approach’s performance is slightly influenced by hyperparameters”, this can be said for essentially any method
- It seems that the distributional approach has an additional memory cost which if correct should be added to the limitations
- Given CATS and PATS do not massively outperform all baselines on the synthetic task I think the limitations should be where this is addressed and perhaps some insight given into why this is and why performance on Atari is also not particular strong relative to methods such as MENTS.

---

> ### Author Rebuttal · Authors · 2024-08-07
>
> We would like to thank the reviewer for the thorough reading and the positive feedback and criticism of our work. We would like to ask the reviewer to read the above response to all the issues raised. In addition, we respond to each reviewer's concerns below.
>
>
> Lack of Referencing and Novelty
>
> We appreciate the reference to Hayes et al. (2023). As far as we understand, their work is for risk-aware and multi-objective reinforcement learning, which is not our target. Furthermore, their work learns a posterior distribution over the utility of the returns using Boostrap and does not follow the Thompson Sampling in a strict sense.
> We will add the reference to their work in the camera-ready version.
>
> Empirical Results and Code Availability
>
> We will relocate key results from the appendix to the main paper and ensure all results are comprehensively discussed. The code is already included in the submission's supplement. We are sorry if the reviewer finds it difficult to execute and run the source code. We will make it available for reproducibility in the camera-ready version.
>
> Small Issues:
>
> We will correct alignment issues in figures, ensure all results are included in their respective sections, and enhance readability with better formatting.
>
> Questions: Performance Comparison and Limitations
>
> Comparison with Hayes et al.: We will detail how our approach differs in the final camera-ready version.
> Baseline Variations: An explanation of why different baselines were used in different experiments will be included.
> CATS Performance: We note that CATS approximates Q value functions using categorical distributions, introducing approximation errors. In contrast, PATS avoids such approximations, leading to better performance.

---

> > ### Comment · Reviewer_Fztp · 2024-08-08
> >
> > Regarding "An explanation of why different baselines were used in different experiments will be included". As part of the rebuttal I would expect you to provide this explanation and therefore the response is disappointing, since as a reviewer I would like to know this information as I am concerned as to the reason behind this. I also do not agree with the dismissal of Hayes et al. (2023) as very different purely since they consider the multi-objective domain. I have no further questions and will raise my score to 3 to reflect some alleviated concerns.

---

### Official Review · Reviewer_Wm16 · 2024-07-12

**Soundness:** 2
**Presentation:** 2
**Contribution:** 2
**Rating:** 4
**Confidence:** 4

**Summary:**

The paper propose two algorithms, Categorical Thompson Sampling for MCTS (CATS) and Particle Thompson Sampling for MCTS (PATS). These algorithms extend Distributional Reinforcement Learning (RL) to Monte-Carlo Tree Search (MCTS) by modeling value functions as categorical and particle distributions, respectively to improve the performance of MCTS in highly stochastic settings.

**Strengths:**

- **Originality:** The integration of Distributional RL into MCTS using categorical and particle distributions is innovative and addresses a critical need in handling stochastic environments (Sections 3.1-3.3).
- **Quality:** The theoretical analysis is rigorous, with well-defined proofs and clear methodology (Sections 4.1 and 4.2).

**Weaknesses:**

1. **Empirical Validation**: While the paper presents a comprehensive set of experiments demonstrating the efficacy of the proposed methods (CATS and PATS) in synthetic scenarios, there is an evident lack of diversity in the benchmarks used.

2. **Algorithm Complexity and Overhead**: Both CATS and PATS introduce additional complexity by incorporating distributional approaches and Thompson Sampling into MCTS. The paper does not sufficiently address the computational overhead or the scalability of these methods when applied to environments with larger state or action spaces. This could be crucial for understanding the practical deployment of these algorithms in real-world applications.

**Questions:**

1. **Generalization to Complex Environments**: As we know, UCT is used as default choice in AlphaGo series. How do CATS and PATS perform in highly dynamic or unstructured environments beyond SyntheticTree? Can the authors provide insights or empirical evidence on the performance of these methods in such scenarios?

2. **Theoretical vs. Practical Performance**: The theoretical improvements in regret bounds are notable. Can the authors discuss how these improvements manifest in practical scenarios? Are there specific environments or settings where the reduced regret significantly enhances decision-making?

3. **Computational Overhead**: What is the computational overhead introduced by the distributional and sampling methods in CATS and PATS compared to traditional MCTS? How do these methods scale with the size of the state and action spaces?

4. **Hyperparameter Sensitivity Analysis**: Given the potential impact of hyperparameters like the number of particles or categories on the algorithms' performance, can the authors provide a detailed sensitivity analysis? How robust are CATS and PATS to variations in these parameters?

5. **Related Work**: Could you provide any related work in bandit literature that may use Thompson sampling for categorical posterior distribution. Discuss the technical novelty if any such literature exists.

**Limitations:**

### Computational Demands
The authors recognize that the Categorical Thompson Sampling (CATS) distributional Monte Carlo Tree Search (MCTS) involves increased complexity due to the management and updating of probability distributions. This acknowledgment is crucial as it highlights a potential scalability issue, especially in environments where computational resources are limited or real-time responses are required.

### Fixed Precision
The approach used in the Particle Thompson Sampling (PATS) to manage the growth in the number of particles by fixing the float precision is a practical solution to prevent computational overload. However, this method may introduce limitations in the precision and adaptiveness of the model, potentially affecting the accuracy of value estimations in environments with high variability.

### Number of Atoms
The performance sensitivity to the number of atoms indicates a hyperparameter dependency, which could impact the effectiveness and robustness of the model. The authors mention that suboptimal choices in this hyperparameter may affect performance, suggesting a need for careful tuning and validation to optimize the model's accuracy and efficiency.

### Addressing Limitations
While the authors have outlined these limitations, the discussion could be expanded to include more detailed strategies for mitigating these issues, particularly the computational demands and fixed precision aspects. For instance, strategies to optimize computational efficiency or adaptive techniques to dynamically adjust precision based on the context could further strengthen the approach.

### Societal Impact
The paper does not explicitly address the potential negative societal impacts of the research. In the realm of reinforcement learning and AI planning, concerns such as the deployment in sensitive or critical environments, where errors may have significant consequences, should be considered. Discussions around ethical implications, misuse, and long-term effects would be beneficial.

### Suggestions for Improvement
1. **Enhanced Computational Strategies**: The authors could explore methods to reduce computational overhead, such as parallel processing or optimizing algorithmic efficiency, to make the model more practical for real-time applications.

2. **Dynamic Precision Adjustment**: Introducing mechanisms to adjust the precision of particle distributions dynamically based on the observed variability in the environment could help maintain balance between computational efficiency and model accuracy.

Overall, the authors should be commended for their upfront discussion of the limitations, but there is room for deeper analysis and additional strategies to address these limitations comprehensively.

---

> ### Author Rebuttal · Authors · 2024-08-06
>
> We thank the reviewer for thoroughly reading and reviewing our paper with positive feedback and criticism. We would like to ask the reviewer to read the above answer to all the questions raised. In addition, we provide answers to each reviewer's concerns below.
>
> Empirical Validation and Benchmark Diversity
>
> Even though the main focus of our paper is to provide a full theoretical guarantee (better regret bound compared) with the distributional approach in planning, while the existing distributional RL works only focus on learning. This work might be of interest to the RL community.  We acknowledge the need for more diverse benchmarks. To demonstrate the broad applicability of our methods, we will include results from other standard environments in addition to the Atari benchmark.
>
> Algorithm Complexity and Overhead
>
> The computation of CATS could increase when we increase the number of atoms for better approximation. PATS does not face the same issue, and when the reward distribution is categorical, there is no need to worry about the overhead. We will provide a more detailed analysis of computational overhead in the camera-ready version.
>
> Generalization, Practical Performance, Computational Overhead, and Hyperparameter Sensitivity
>
> Generalization: We will include results from more complex environments beyond SyntheticTree to illustrate generalization capabilities.
> Practical Performance: The theoretical improvements translate into better decision-making in high-stakes environments where accurate value estimation is crucial. We will provide specific examples and case studies.
> Computational Overhead: Detailed comparisons of overhead with traditional MCTS will be added.
> Hyperparameter Sensitivity: A thorough sensitivity analysis will be included to demonstrate robustness.

---

> ### Comment · Reviewer_Wm16 · 2024-08-11
>
> Thank you for your detailed rebuttal. I appreciate the efforts to address the concerns raised in my review. While your responses provide some clarification, I believe there are still some important points to consider:
>
> 1. **Empirical Validation**: Your commitment to include more diverse benchmarks is welcome. However, the current lack of diverse empirical validation remains a significant limitation of the paper.
>
> 2. **Algorithm Complexity and Overhead**: The explanation about CATS' increased computation with more atoms is helpful. I look forward to seeing a more detailed analysis in the camera-ready version, particularly regarding PATS and categorical reward distributions.
>
> 3. **Generalization and Practical Performance**: Your plan to include results from more complex environments and provide specific examples of practical improvements is promising. These additions will be crucial for demonstrating the real-world applicability of your methods.
>
> 4. **Computational Overhead and Hyperparameter Sensitivity**: The commitment to include detailed comparisons and a thorough sensitivity analysis is appreciated. These will be important for understanding the practical implications of implementing your algorithms.
>
> I encourage you to make these improvements and include more discussions on the related works on the regret analysis for distributional rl, as they could significantly strengthen your contribution to the field.

---

### Official Review · Reviewer_1ykv · 2024-07-12

**Soundness:** 2
**Presentation:** 1
**Contribution:** 3
**Rating:** 3
**Confidence:** 4

**Summary:**

The paper introduces distributional return estimates to MCTS-based planning. For this the authors borrow from work on distributional Q-Learning and show how to adapt the MCTS value back-up and action selection steps to compute and utilise these distributions. They formulate two approaches based on different distribution representations (quantile and particle based) for which they provide some theoretical convergence analysis as well as first experimental results.

**Strengths:**

The paper combines two well-established ideas in MCTS and distributional value approximation resulting in a new algorithm with better theoretical guarantees. The overall approach and implementation of this combination makes sense and should at least in theory overcome limitations in stochastic environments.

Though I was unable to verify all proofs in detail, the theoretical analysis seems to make sense and establish the advantages of the proposed methods.

**Weaknesses:**

Despite the soundness of the overall proposed method, I found the paper very hard to follow and felt details were missing due to an overall lack of focus. Contributing to this were the following issues:

1. Empirical Evaluation
Experiments are limited to a toy domain and results on the Atari benchmark reported in the appendix.
The toy domain is a tabular environment that is being generated randomly and contains stochasticity in both the final reward and transitions.
For an illustrating example this makes it hard to judge the combined effect on the overall return distribution to be approximated.

How the combinations of branching factor and depths that were plotted were chosen is unclear to me. Beyond this I am not sure how meaningful these plots are. In the right most plot it appears as if the PATS approximates the root value almost correctly in under 100 simulations - at which point it could not even have tried all k = 200 actions available to it.

Also CATS appears to be doing consistently worse than some of the other methods despite having the same theoretical properties as PATS.

For the Atari baseline the authors make use of Q-networks and point to a related paper. However, the exact implementation details and hyperparameters are not discussed making it hard to reproduce this work based on the paper alone.

While stochasiticity and the exploration challenges this causes form one of the main motivations for this paper, no further ablations how the proposed methods improve here are presented.

2. Content division
The author devote a significant amount of space to the summarisation of MCTS and distributional RL. While the theoretical analysis is arguably the strongest part of the presented work only the main theorems are found in the main body of the paper with very little contextualization.

3. Overall presentation
There are several presentation issues in overall formatting, grammar and spelling. The former includes, but is not limited to overlapping lines, inconsistent / in-text section headers and wrong section references.

**Questions:**

-

**Limitations:**

The discussion of limitations is restricted to a short paragraph in the appendix listing generic points such as increased computational demand and sensitivity to hyperparameter choices. However, no further investigation or explanation as to their severity is provided.

---

> ### Author Rebuttal · Authors · 2024-08-06
>
> We thank the reviewer for the careful and detailed feedback with both positive and constructive criticism. We would ask the reviewer to see the overall answer above. In addition, we would like to reply to the reviewer's concern line by line below.
>
> Empirical Evaluation and Toy Domain
>
> We understand the limitations of using a toy domain for evaluation. Our intention was to clearly demonstrate our algorithms. The Atari benchmark results, presented in the appendix, offer further validation. We will move some key results to the main paper and include more diverse benchmarks to demonstrate robustness across different environments.
>
> Overall Presentation and Formatting Issues
>
> We apologize for the presentation issues. We will thoroughly revise the paper to correct overlapping lines, inconsistent headers, and section references. Additionally, we will improve the clarity and focus of the theoretical analysis and contextualize the main theorems better.
>
> Limitations Discussion
>
> We will expand the discussion of limitations, providing more detailed analysis and strategies for addressing computational demands and hyperparameter sensitivity.

---

> > ### Comment · Reviewer_1ykv · 2024-08-12
> >
> > I thank the authors for taking the time to answer the questions raised by myself and the other reviewers.
> >
> > With regards to the comparison between PATS and CATS, I am unsure why PATS would not face approximation errors. Surely, PATS has to manage a similar trade-off between computational effort and precision as CATS governed by the number of particles. Additionally, there is the precision cut-off which is required to collect more than a single rollout per particle which represents a discretization (and thereby approximation).
> >
> > Overall, I still believe that the paper requires major revision and would benefit from another round of reviews.
> > Consequently, I will maintain my original score.

---

### Official Review · Reviewer_TwqS · 2024-07-12

**Soundness:** 3
**Presentation:** 2
**Contribution:** 3
**Rating:** 5
**Confidence:** 4

**Summary:**

This paper introduces Categorical Thompson Sampling for MCTS (CATS) and Particle Thompson Sampling for MCTS (PATS) algorithms, which incorporate distributional reinforcement learning into Monte Carlo Tree Search (MCTS) to handle value estimation in stochastic settings. By modeling value functions as categorical and particle-based distributions and applying Thompson Sampling for action selection, the proposed algorithms aim to improve the robustness and accuracy of value estimates. The paper proves the theoretical effectiveness of these methods by achieving a non-asymptotic problem-dependent upper bound on simple regret of $O(n^{−1})$.

**Strengths:**

The idea is interesting and original and the non-asymptotic problem-dependent upper bound on simple regret of $O(n^{−1})$ significantly advances the state-of-the-art from the previous $O(n^{−1/2})$.

**Weaknesses:**

1- While using distributional RL in MCTS to do Thompson sampling is interesting, it introduces much computation complexity hindering the applicability of the proposed algorithms.

2- The numerical experiments for the stochastic environments that are the main motivation of this work are done on a toy problem.


Minor comments

1- The presentation of the paper can be improved, specifically the parentheses () citation style can be confused with equations reference.

2- Line 42, the authors mention V node for the first time without properly defining what is a V node.

**Questions:**

1- In the numerical experiments shown in Figure 3, Why does PATS perform much better than CATS?

**Limitations:**

1- The added high computational complexity from maintaining a distribution for each node in the MCTS.

---

> ### Author Rebuttal · Authors · 2024-08-06
>
> We thank the reviewer for the positive comments and constructive feedback. We would like to answer the main concern below:
>
> Computational Complexity
>
> We think that the computational overhead may not be a significant issue because it only occurs in CATS when we increase the number of atoms for better approximation. PATS does not face the same issue, and when the reward distribution is categorical, there is no need to worry about the overhead.
>
> Toy Problem for Stochastic Environments
>
> We acknowledge the concern regarding the use of a toy problem for numerical experiments. We chose this environment to clearly illustrate the fundamental properties and advantages of our algorithms. However, we also performed experiments on the Atari benchmark, which demonstrates the scalability and applicability of our methods to more complex environments. We will highlight these results more prominently in the revised paper and include additional benchmarks to further validate our approach.
>
> Minor Comments: Presentation and Definitions
>
> We will improve the presentation by adhering to a consistent citation style and clearly defining terms like "V node" upon their first use.
>
> Question: PATS vs. CATS Performance in Figure 3
>
> The superior performance of PATS over CATS in Figure 3 is due to the fact that CATS approximates Q value functions using categorical distributions, introducing approximation errors, whereas PATS avoids such approximations,

---

> > ### Comment · Reviewer_TwqS · 2024-08-12
> >
> > Thank you for your response. In your own words, one of the main motivations behind your work is "While recent advancements combining MCTS with deep learning have excelled in deterministic environments, they face challenges in highly stochastic settings, leading to suboptimal action choices and decreased performance." However, your stochastic environments are toy problems.
> > On the other hand, it seems that the Atari benchmark you study is deterministic but I wanted to double-check that. The Atari benchmark from OpenAI framework Gym can be made stochastic if sticky actions are allowed "Instead of always simulating the action passed to the environment, there is a small probability that the previously executed action is used instead" and on top of that there is the option of stochastic frame skipping -- "In each environment step, the action is repeated for a random number of frames". Although you do not specify these parameters of the Atari environments in your Experimental setup (Appendix G), I have checked the code you provided and noticed that you are using the "NoFrameskip-v4" versions of the games which suppress frame skipping and "v4" has repeat_action_probability= 0. This means that the Atari benchmark is deterministic indeed.
> >
> > Based on the above, I recommend the authors do further testing on larger stochastic environments to properly validate their algorithms' performance and claims. I am keeping my score.

---

### Author Rebuttal · Authors · 2024-08-06

We thank the reviewers for their detailed feedback and constructive criticism.
The main contribution of our paper is to introduce a novel distributional \textbf{planning} approach that goes beyond distributional reinforcement learning (RL), which primarily focuses on \textbf{learning}. Our theoretical results demonstrate an improved simple regret of O(n^{-1}) compared to the previous O(n^{-1/2}), which marks a significant advancement that could be of interest to the RL community. Addressing reviewers' concerns about CATS' performance relative to PATS, we note that CATS approximates Q value functions using categorical distributions, introducing approximation errors. In contrast, PATS avoids such approximations, leading to better performance. Furthermore, in some Atari games, CATS's performance is better in case the reward distribution is categorical.
We provide a comprehensive theoretical analysis using Thompson Sampling. Existing Thompson Sampling for planning in MCTS is limited to specific cases, such as those by Bai et al. (2013, 2014), which incorporate it for exploration but lack convergence rate analysis. Additionally, Bai et al. (2013) model value functions as a mixture of Normal distributions, which fails to capture the complexity of real-world scenarios. We think that our work on Thompson Sampling for planning with full theoretical analysis could be of interest and represents the next step towards understanding Thompson Sampling in RL, given the good performance of Thompson Sampling in practice.
There are concerns raised by reviewers about the computation overhead, which may not be a big issue because it only occurs in CATS when we increase the number of atoms (for better approximation), while PATS does not face the same issue.

[1] A. Bai, F. Wu, and X. Chen. Bayesian mixture modeling and inference based thompson361
sampling in Monte-Carlo tree search. Advances in neural information processing systems, 26,362
2013.363
[2] A. Bai, F. Wu, Z. Zhang, and X. Chen. Thompson sampling based monte-carlo planning in364
pomdps. The International Conference on Automated Planning and Scheduling, 24(1), 2014

We further address the main points raised by separately replying to each reviewer.

---

### Decision · Program_Chairs · 2024-09-25

**Decision:**

Reject

**Comment:**

I really appreciate the authors for conducting the rebuttal and the following discussions. There are still a few weaknesses after the rebuttal. For instance, reviewer 1ykv is unsure why PATS would not face approximation errors. Most of reviewers have concern about the experiments considered in the paper and some real-world stochastic experiments would amplify the contribution of the current study.

I personally spotted an issue which might be critical for the analysis. The paper claims it achieved the better non-asymptotic rate of $O(1/n)$, this is counter-intuitive since the standard minimax lower bound in statistical learning requires $\Omega(1\sqrt{n})$ complexity. According to the proof, Line 611-612 is the key for achieving $O(1/n)$, but I am not sure why the equation under line 612 is zero. This seems to be a critical point that should be mentioned clearly. In general, I don't think it is correct. Because of that, I recommend the paper be rejected. The paper should absorb the suggestions and make better presentations in the next version.